# Harnessing peak transmission around symptom onset for non-pharmaceutical intervention and containment of the COVID-19 pandemic

Liang Tian [1,2,10], Xuefei Li [1,3,10], Fei Qi [1,3], Qian-Yuan Tang [1,4], Viola Tang [1,5], Jiang Liu [1], Zhiyuan Li [1,6], Xingye Cheng [1,2], Xuanxuan Li[1,7,9], Yingchen Shi [1,7,8], Haiguang Liu [1,8,9] & Lei-Han Tang [1,2,8 ✉]

Within a short period of time, COVID-19 grew into a world-wide pandemic. Transmission by pre-symptomatic and asymptomatic viral carriers rendered intervention and containment of the disease extremely challenging. Based on reported infection case studies, we construct an epidemiological model that focuses on transmission around the symptom onset. The model is calibrated against incubation period and pairwise transmission statistics during the initial outbreaks of the pandemic outside Wuhan with minimal non-pharmaceutical interventions. Mathematical treatment of the model yields explicit expressions for the size of latent and pre-symptomatic subpopulations during the exponential growth phase, with the local epidemic growth rate as input. We then explore reduction of the basic reproduction number $R_0$ through specific transmission control measures such as contact tracing, testing, social distancing, wearing masks and sheltering in place. When these measures are implemented in combination, their effects on $R_0$ multiply. We also compare our model behaviour to the first wave of the COVID-19 spreading in various affected regions and highlight generic and less generic features of the pandemic development.

[1] COVID-19 Modelling Group, Hong Kong Baptist University, Kowloon, Hong Kong SAR, China. [2] Department of Physics and Institute of Computational and Theoretical Studies, Hong Kong Baptist University, Kowloon, Hong Kong SAR, China. [3] CAS Key Laboratory of Quantitative Engineering Biology, Shenzhen Institute of Synthetic Biology, Shenzhen Institutes of Advanced Technology, Shenzhen, China. [4] Center for Complex Systems Biology, Universal Biology Institute, University of Tokyo, Tokyo, Japan. [5] Department of Information Systems, Business Statistics and Operations Management, Hong Kong University of Science and Technology, Hong Kong SAR, China. [6] Center for Quantitative Biology, Peking University, Haidian, Beijing, China. [7] Department of Engineering Physics, Tsinghua University, Haidian, Beijing, China. [8] Complex Systems Division, Beijing Computational Science Research Center, Haidian, Beijing, China. [9] Physics Department, Beijing Normal University, Haidian, Beijing, China. [10] These authors contributed equally: Liang Tian, Xuefei Li. ✉email: lhtang@hkbu.edu.hk

The Coronavirus Disease 2019 (COVID-19) is a new contagious disease caused by the novel coronavirus (SARS-COV-2)[1], which belongs to the genera of *betacoronavirus*, the same as the coronavirus that caused the SARS epidemic between 2002 and 2003[2]. COVID-19 has spread to more than 200 countries/regions, with over 102 million confirmed cases and 2.2 million lives claimed as of January 31, 2021[3]. The outbreak has been declared a pandemic and a public health emergency of international concern[4].

As the specific symptoms of COVID-19 are now well-publicised, symptomatic transmissions are being contained in most countries. However, disease transmission by pre-symptomatic and asymptomatic viral carriers is seen to be extremely difficult to deal with due to its hidden nature[5]. Clinical data reveals that viral load becomes significant before the symptom onset[6–8]. Epidemiological investigations have identified clear cases of pre-symptomatic transmission soon after the initial outbreak[9–12]. Estimates vary greatly among experts on the percentage of total transmission due to this group of viral carriers, ranging from as low as 18% to over 50%[13–15]. An early model-based study by Ferretti et al.[16] suggested that pre-symptomatic transmission alone could yield a basic reproduction number $R_{0,\mathrm{p}} = 0.9$, close to the critical value of 1.0 that sustains epidemic growth. Under intense surveillance of the pandemic, pre-symptomatic and asymptomatic transmissions become the main focus in outbreak control[5].

While the actual viral shedding is influenced by many factors, patient viral load during the course of disease progression is more universal. This suggests a modelling approach that starts with clinical observations of symptom onset, and treats disease transmission as a dependent process that is further shaped by living and social conditions, including control measures to reduce physical contact. Following this strategy, we first introduce a model for an unprotected population and calibrate the model parameters against clinical case reports during the initial outbreak. Subsequently, we estimate the percentage reduction in the basic reproduction number (estimated to be around 3.87 at an exponential growth rate of 0.3/day) due to contact tracing, mask wearing and other measures, individually or in combination. Additionally, we present our findings against the epidemic development curves around the world to highlight the level of social mobilisation required to contain COVID-19 spreading.

## Results

**A renewal process centred on symptom onset.** In epidemiological studies, the central quantity is the average number of secondary infections per unit time $r(t)$ by a viral carrier on day $t$ since the individual's own infection[17,18]. In the case of COVID-19, disease transmission peaks around the symptom onset time of the individual[7,8], as illustrated by the infectiousness curve shown in Fig. 1a (left panel). This property, when averaged over the population, gives an $r(t)$ (Fig. 1a, right panel) that closely resembles the symptom onset time distribution, which we denote by $p_\mathrm{O}(t)$ (Fig. 1a, middle panel). In fact, when the time window of transmission is narrowly centred around the symptom onset, we have approximately

$$r(t) \approx R_\mathrm{E} p_\mathrm{O}\big(t + \theta_\mathrm{S}\big). \tag{1}$$

The mean reproduction number $R_\mathrm{E}$ sets the overall level of disease transmission in the population, and equals the basic reproduction number $R_0$ when the infectious disease first breaks into a community. Its actual value could change over time due to factors such as the intervention and containment measures considered below. The shift parameter $\theta_\mathrm{S}$ (Fig. 1a, right panel)

accommodates the actual shape of the infectiousness curve as well as effects resulting from intervention measures, e.g., isolation delays of infected cases.

To link up Eq. (1) with actual transmission data, we developed a compartmentalised epidemic spreading model as illustrated in Fig. 1b. A total of four phases are introduced to accommodate the infectiousness curve in Fig. 1a, left panel. Three of these phases reside in the pre-symptomatic period: a non-infectious latent phase L, followed by infectious phases $A_1$ and $A_2$ before and after the infectiousness peak. Starting from the day of infection, an individual first stays in the latent phase L. Transition to phase $A_1$ takes place at a rate $\alpha_\mathrm{L}(t)$ that depends on the elapsed time $t$ since infection. Once in phase $A_1$, the individual is infectious with a daily transmission rate $\beta_\mathrm{A}$. Duration of the $A_1$ phase is variable and follows Poisson statistics with an exit rate constant $\alpha_\mathrm{A}$. On the other hand, duration of the succeeding phase $A_2$ is fixed at $\theta_\mathrm{P}$, after which symptoms develop and the person enters the symptomatic phase S. Upon entering $A_2$, the patient's disease transmission rate $\beta_\mathrm{B}(\tau)$ weakens with the elapsed time $\tau = t - t_\mathrm{O} + \theta_\mathrm{P}$ to match the right-wing of the infectiousness curve. Note that, due to the variable duration of $A_1$, the population-averaged infectiousness of this phase rises towards the symptom onset.

Applying the above rules of disease transmission to a large and well-mixed population, the number of new infections per unit time $J_\mathrm{L}(T)$ on day $T$ satisfies the renewal equation

$$J_\mathrm{L}(T) = \int_0^\infty r(t) J_\mathrm{L}(T - t) \mathrm{d}t, \tag{2}$$

where the kernel function is given by

$$r(t) = \int_0^t \alpha_\mathrm{L}(t_1) q_\mathrm{L}(t_1) e^{-\alpha_\mathrm{A}(t - t_1)} \left[ \beta_\mathrm{A} + \alpha_\mathrm{A} \int_0^{t - t_1} \beta_\mathrm{B}(t_2) e^{\alpha_\mathrm{A} t_2} \mathrm{d}t_2 \right] \mathrm{d}t_1, \tag{3}$$

with $q_\mathrm{L}(t) = e^{-\int_0^t \alpha_\mathrm{L}(t_1) \mathrm{d}t_1}$ being the probability that an individual remains in the latent phase $t$ days after infection. Derivation of these results are presented in Supplementary Section 1, together with dynamic equations governing the size of each subgroup.

Equations (2) and (3) can be solved by performing the Laplace transform. In this respect our model is equally tractable mathematically as the susceptible-exposed-infectious-recovered (SEIR) type models defined by a set of rate equations[19]. As we show below, the explicit representation of the temporal structure for disease progression and transmission in the present case facilitates direct model calibration from clinical data and also quantitative evaluation of intervention measures against epidemic development.

In Supplementary Section 1.3, we show that the mean reproduction number of the model is given by $R_\mathrm{E} = R_\mathrm{E}^\mathrm{A} + R_\mathrm{E}^\mathrm{S}$, with $R_\mathrm{E}^\mathrm{A} = \beta_\mathrm{A}/\alpha_\mathrm{A} + \int_0^{\theta_\mathrm{P}} \beta_\mathrm{B}(\tau) \mathrm{d}\tau$ and $R_\mathrm{E}^\mathrm{S} = \int_{\theta_\mathrm{P}}^\infty \beta_\mathrm{B}(\tau) \mathrm{d}\tau$ being reproduction numbers associated with pre-symptomatic and symptomatic transmissions, respectively. When the right wing of the infectiousness curve in Fig. 1a takes the form of an exponentially decaying function $\beta_\mathrm{B}(\tau) = \beta_\mathrm{A} e^{-\alpha_\mathrm{B} \tau}$ with a sufficiently large decay rate $\alpha_\mathrm{B}$, we recover Eq. (1) which was initially proposed on heuristic grounds. The shift parameter is given approximately by

$$\theta_\mathrm{S} \approx \theta_\mathrm{P} - \frac{\alpha_\mathrm{A}}{\alpha_\mathrm{B}(\alpha_\mathrm{A} + \alpha_\mathrm{B})}. \tag{4}$$

**Parameter calibration.** By combining three data sets[11,20,21] with a total of 347 infection cases outside the Hubei province in China, we estimated the incubation period statistics $p_\mathrm{O}(t)$ (see Fig. 2a). Due to the difficulty in identifying a precise date of infection, a

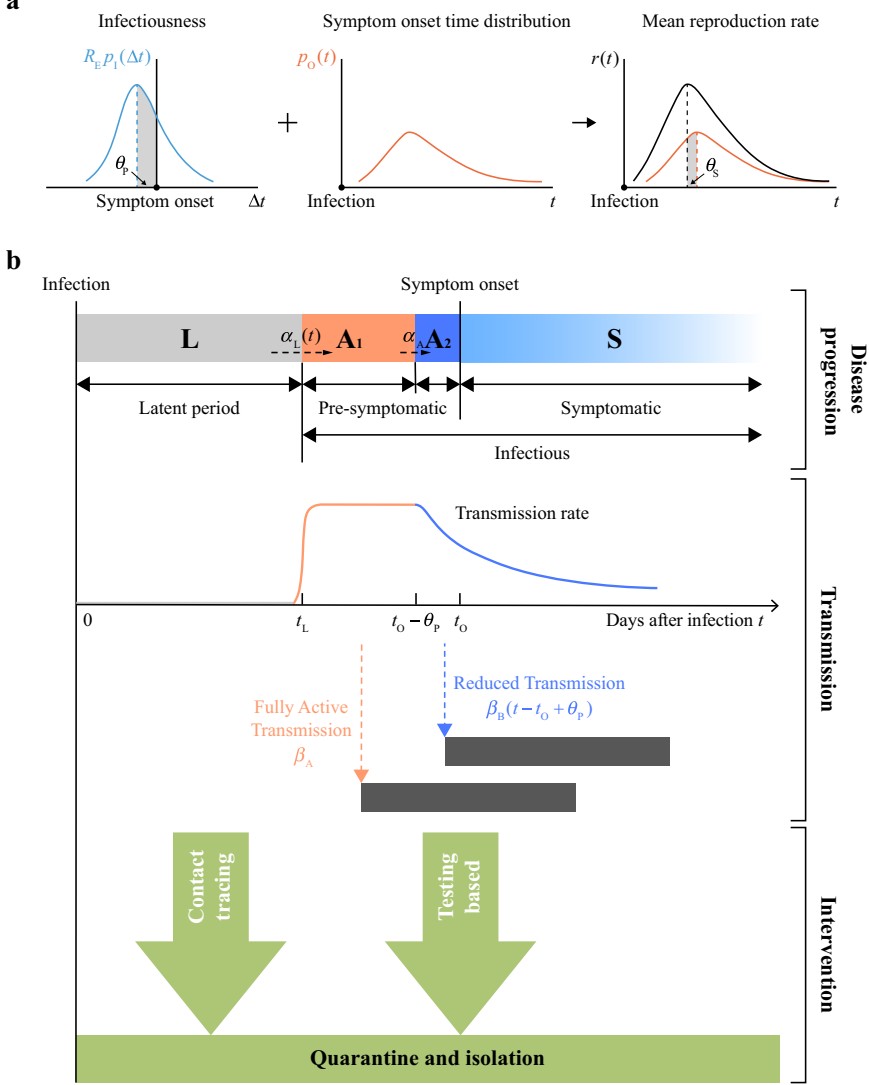

**Fig. 1 A stochastic model for COVID-19 disease progression, transmission and intervention. a** The mean reproduction rate $r(t)$ (black curve) of a patient on day $t$ since infection is expressed as a convolution of the symptom onset time distribution $p_O(t)$ (red curve) and the infectiousness curve $R_E p_I(\Delta t)$ (blue curve), where $\Delta t$ is measured from the symptom onset. The mean reproduction number $R_E$ sets the overall level of the epidemic. The peak of the normalised infectiousness function $p_I(\Delta t)$ is shifted from the symptom onset by an amount $\theta_P$, which takes a positive value on the pre-symptomatic side. The peak of the mean reproduction rate $r(t)$ is shifted from the peak of the symptom onset time distribution $p_O(t)$ by $\theta_S$. **b** A compartmentalised model. A person infected first goes through a non-infectious latent phase (L) until $t_L$, followed by an infectious period that spans across symptom onset at $t_O$. In the pre-symptomatic phase A, the person is infectious without symptoms. The A phase is further split into two subphases, $A_1$ with a constant transmission rate (orange region) and $A_2$ with a declining transmission rate (blue region). At the symptom onset time $t_O$, the person enters the S phase, and continues to be infectious (light blue region). Contact tracing brings an infected person out of the transmission cycle at the point of isolation, while testing does so only when the result is positive.

window is assigned to the incubation period in each case. A rudimentary way to deal with the uncertainty is to treat all possible values inside the window as equally likely. This procedure yields a statistical distribution for each of the three data sets as well as the conglomerated one, as shown by symbols in Fig. 2a.

Alternatively, viewing the data as samples of a common underlying probability distribution, we estimated $p_O(t)$ by likelihood maximisation (see the "Methods" section and Supplementary Section 2.1). Within the class of functions considered, the log-normal distribution combined with an exponential tail yields the largest likelihood value (Fig. 2a, red line). From day 6 onward, $p_O(t)$ follows an exponential decay with a rate of $-0.31$/day, with a 95% confidence interval (CI) of $(-0.35, -0.27)$ per day. We have

also examined other values (from day 4 to day 8) for the switch. In all cases, exponential tail decay rates are found to be round $-0.31$/day (see Supplementary Section 2.1).

A data set of 77 pairwise transmissions in several eastern and southeastern Asian countries and regions during their initial COVID-19 outbreak was compiled by He et al.[7] We took 66 pairs with unique symptom onset of the primary case to estimate the transmission probability $p_I(\Delta t)$, with $\Delta t$ measured from the symptom onset of the primary case. The results are shown in Fig. 2b (see the "Methods" section and Supplementary Section 2.2 for details). Under a maximum-likelihood estimation scheme, we considered three alternative forms for $p_I(\Delta t)$. All have exponential tails far away from the transmission peak, but differ in the way

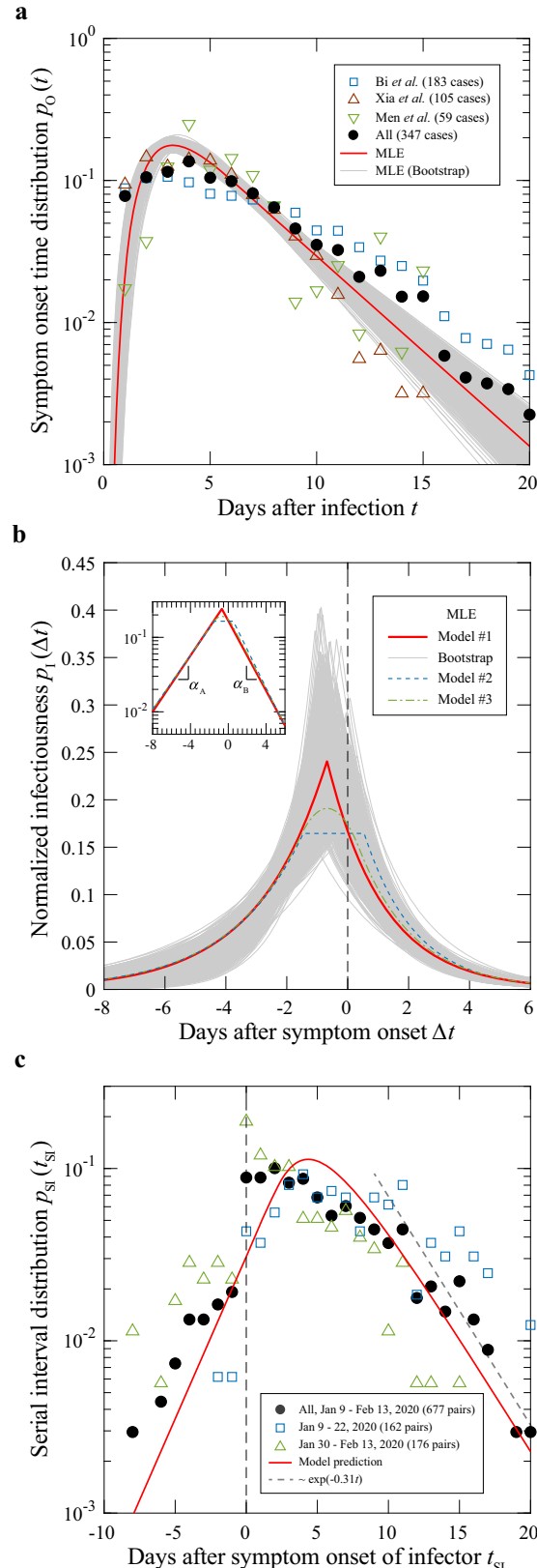

**a**

**b**

**c**

**Fig. 2 Parameter calibration from case studies. a** The symptom onset time distribution. Raw statistics of three reported data sets (up triangle[11], down triangle[20] and square[21]) and their union (solid circle) are shown. The red curve gives the estimated distribution under a maximum-likelihood scheme. Grey thin curves are generated with bootstrapping (see the "Methods" section). MLE maximum-likelihood estimation. **b** The infectiousness function. The data set contains 66 transmission pairs[7]. Result of the MLE is given by two exponential functions meeting at −0.68 days (red solid line). Also shown are distributions with a constant bridge (dash line), or with a dome cap (dash-dotted line), with slightly lower likelihood values (see Supplementary Section 2.2). Thin grey curves are from bootstrapping for model #1 (see the "Methods" section). **c** Serial interval statistics outside the Hubei province in China from January 9 to February 13, 2020[22,23]: whole period (solid circles), first two weeks (open squares), and last two weeks (open triangles). The grey dashed line on the right indicates exponential decay at a rate −0.31/day. The red curve is the convolution of the two red curves shown in **a** and **b**.

cusp function, with its peak located at 0.68 days before the symptom onset, is the most probable for this data set (Fig. 2b). Decay rates for the left and right wings are given by 0.46/day and 0.54/day, respectively (see Supplementary Section 2.2.2).

Duration of the $A_1$ phase follows a Poisson process and is hence exponentially distributed. This gives rise to an exponential tail of the population-averaged infectiousness curve prior to entering the $A_2$ phase. We therefore set the model parameters to $\alpha_A = 0.46$/day, $\theta_P = 0.68$ days, and $\beta_B(\tau) = \beta_A e^{-\alpha_B \tau}$ with $\alpha_B = 0.54$/day. These values were used in our numerical calculations, with the corresponding CIs given in Table 1.

**Serial interval.** Xu et al. [22] compiled a database of 1407 COVID-19 transmission pairs outside the Hubei province in China between early January till mid-February 2020. Among them, 677 pairs have the symptom onset dates and social relationships of infector–infectees. A detailed analysis of the data set, stratified before, during and one week after the Wuhan lockdown on 23 January 2020, was carried out by Ali et al. [23] which showed reduction of the serial interval of symptom onsets by a factor of 3 over the 5 weeks. In Fig. 2c, we show the distribution of the serial interval data for the whole period (solid circles) and separately for the first (open squares) and last two weeks (open triangles) of the period. The red line gives the predicted serial interval distribution

$$p_{SI}(t_{SI}) = \int_{-\infty}^{t_{SI}} p_I(\Delta t) p_O(t_{SI} - \Delta t) \, d\Delta t, \quad (5)$$

using our estimated values for $p_O(t)$ and $p_I(\Delta t)$ (see Supplementary Section 2). While the overall agreement with the unstratified data is good especially on the positive side, it is also evident that serial intervals can be affected by travel and prevention measures, such as the percentage of imported cases among the infectors, the typical length of isolation delays, etc., which changed substantially before and after the Wuhan lockdown. Such temporal effect on the serial interval can be simulated simply with a shape function that masks $p_I(\Delta t)$. For example, imported index cases who spent part of their infectious period outside the region shift $p_{SI}(t_{SI})$ to the right. On the other hand, vigorous contact tracing shortens isolation delays significantly, which in turn shifts $p_{SI}(t_{SI})$ to the left.

The long-time tail of both $p_O(t)$ and $p_{SI}(t_{SI})$ decays slower than the rates $\alpha_A$ and $\alpha_B$ associated with the infectiousness curve. We have computed the tail of the probability $q_L(t)$ to remain in the latent phase, whose decay rate matches that of $p_O(t)$ (see Supplementary Section 2.4). According to Eq. (1), the long-time

the two wings are joined together in the peak region. In the first case, the two exponential tails join directly to produce a cusp in the middle. In the second case, a flat top of variable width is introduced. In the third case, the flat top is replaced by a parabolic cap to give a more rounded peak. It turns out that the

**Table 1 Key variables and parameters of the compartmentalised model.**

| Size variable | Subpopulation | |
|---|---|---|
| L | Latent, infected but not infectious | |
| $A_1$ | Pre-symptomatic and infectious, constant transmission rate | |
| $A_2$ | Pre-symptomatic and infectious, decreasing transmission rate | |
| S | Symptomatic and infectious, diminishing transmission rate | |
| **Epidemic characteristic** | **Definition (unit)** | |
| $R_E$ | Mean reproduction number of an infected individual | |
| $\lambda$ | Exponential growth rate (per day) | |
| $r(t)$ | Mean reproduction rate on day $t$ since infection (per day) | |
| $p_O(t)$ | Symptom onset time/Incubation period distribution | |
| $p_I(\Delta t)$ | Distribution of infection time $\Delta t$ in a transmission pair, measured from the symptom onset of the index patient | |
| $p_{SI}(t_{SI})$ | Distribution of the delay time in the symptom onset of a transmission pair | |
| **Parameter** | **Definition (unit)** | **Estimated value (95% CI)** |
| $t_O$ | Symptom onset time/Incubation period (days) | Mean ($\tau_O$): 6.04 (5.70, 6.37)[a] |
| | | Median: 4.60 (4.33, 4.88)[b] |
| | | Variance ($\sigma_O$): 4.11 (3.77, 4.46) |
| $\gamma$ | Exponential decay rate of $p_O(t)$ after 6 days (per day) | 0.31 (0.27, 0.35) |
| $\alpha_L(t)$ | Transition rate from L to $A_1$ (per day) | Calibrated through $p_O(t)$ (see Supplementary Section 2.4 for details) |
| $\beta_A$ | Transmission rate in $A_1$ phase (per day) | 0.97 (0.74, 1.27) at epidemic daily growth rate $\lambda = 0.3$/day |
| $\alpha_A$ | Transition rate from $A_1$ to $A_2$ (per day) | 0.43 (0.32, 0.69) |
| $\beta_B(t_B)$ | Transmission rate in $A_2 + S$ (per day) | $\beta_B(t_B) = \beta_A e^{-\alpha_B t_B}$ |
| $\alpha_B$ | Decay rate of infectiousness in $A_2 + S$ (per day) | 0.54 (0.48, 0.65) |
| $R_0$ | Basic reproduction number | 3.87 (3.38, 4.48) at epidemic growth rate $\lambda = 0.3$/day |
| $\theta_P$ | Duration of $A_2$ phase (days) | 0.68 (0.12, 1.02) |
| $\theta_S$ | Reproduction shift parameter (days) | $-0.15$ ($-0.60$, 0.23) |

[a]Mean incubation period was estimated at 5.95 days (95% CI 4.94, 7.11) and 6.4 days (95% CI 5.6, 7.7), respectively[21,45].
[b]Median incubation period was estimated at 5.1 days (95% CI 4.5, 5.8)[46].

tail of the mean reproduction rate $r(t)$ can also be attributed to infected cases who have a long incubation period in their disease progression.

**Mean reproduction number.** Under Eq. (1), the well-known Lotka–Euler estimating equation[24] yields

$$R_E = \frac{e^{-\lambda \theta_S}}{\tilde{p}_O(\lambda)}, \qquad (6)$$

where $\tilde{p}_O(\lambda) = \int_0^\infty p_O(t)e^{-\lambda t}dt$ is the Laplace transform of $p_O(t)$ (see Supplementary Sections 1.3 and 3.1). Using the estimated values above, we obtain from Eq. (6) the $R_E$ versus $\lambda$ curve shown in Fig. 3a, which covers both the growth ($\lambda > 0$) and declining ($\lambda < 0$) phases of the epidemic. The slope of the curve at $R_E = 1$ is given by $1/\tau_g$, where $\tau_g$ is the mean generation time and equals $\tau_O - \theta_S = 6.19$ days under Eq. (6). The intercept of the curve at $R_E = 0$ gives an ultimate epidemic decay rate of $-0.31$/day when disease transmission comes to a complete halt.

To estimate the uncertainty in the computed $R_E - \lambda$ curve, we performed bootstrap analysis of the data used to obtain $p_O(t)$ and $p_I(\Delta t)$. The detailed procedure is described in Supplementary Section 2, with the result shown in Fig. 3a. At a growth rate of $\lambda = 0.3$/day, our estimated value for the basic reproduction number $R_0$ is 3.87 (95% CI [3.38, 4.48]).

**Composition of the infected population.** As we demonstrate in the Supplementary Information, the convolutional form of our main Eq. (2) enables many analytic results to be derived and evaluated with the calibrated parameters. Figure 3b shows the probabilities that a given individual is in one of the four phases on day $t$ after infection, computed using the formula in Supplementary Table 1. The red line marks the boundary between the pre-symptomatic and symptomatic phases. The width of the

orange-coloured region ($A_1$ phase), on the other hand, is proportional to $\alpha_A^{-1} \approx 2$ days.

Figure 3c, obtained from the Laplace transforms of these curves, gives the percentage of the infected population in each of the four phases on a given day when the epidemic is growing at a rate $\lambda$. These curves allow for estimation of the hidden population in L, $A_1$ and $A_2$ phases from the knowledge of S in real-time. They form the basis for quantitative assessment of intervention measures discussed below. Note that at high growth rates, a larger percentage of the infected population is in the latent and pre-symptomatic phases, so that suppressing transmission by this group through, say mask wearing and social distancing, assumes a greater priority.

**Testing and contact tracing.** To break the transmission chain in the community, governments around the world have adopted two measures with varying levels of intensity: (1) testing and isolating infected individuals and (2) tracing and quarantining contacts of infected individuals.

For testing control, persons who were in close proximity to a confirmed infection case are asked to undergo voluntary or mandatory testing for infection, and quarantined when the result is positive. From Fig. 3b we see that, if the test is conducted shortly after infection, the individual has a high probability to still be in the latent phase, hence the test result is likely to be negative. On the other hand, if the test is conducted too late, the person may have already infected others so that the reduction of $r(t)$ given by Eq. (1) is small. Therefore, there is an optimal window between the infection date and the test date, which we analyse in Supplementary Section 4.1.2. In Fig. 4a, we show the reduction of the basic reproduction number $R_0$ as a function of the reporting delay, assuming all suspected contacts are tested. At $R_0 = 3.87$, if the results become available immediately after testing, the reduction of $R_0$ is shown as the blue curve, better than the

**a**

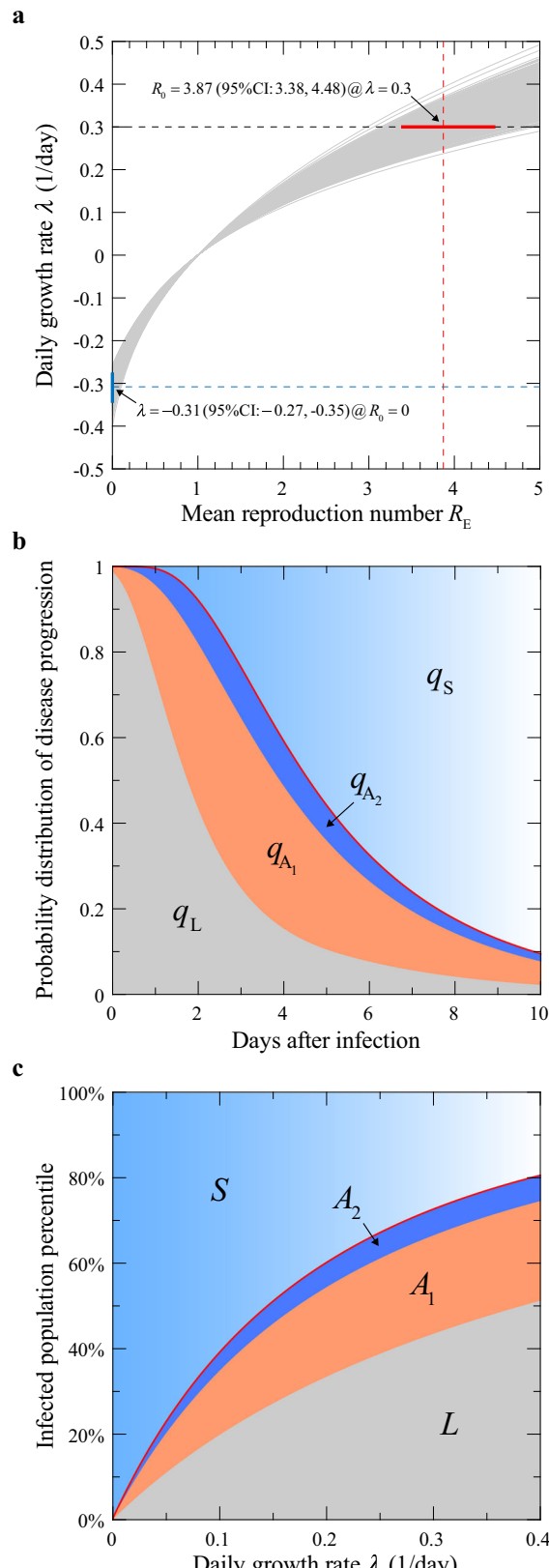

**Fig. 3 Basic model predictions. a** The relationship between the epidemic growth rate $\lambda$ and the mean reproduction number $R_E$. The grey lines, generated using the data shown in Fig. 2 with bootstrapping, give the range of uncertainty in the estimated function. At $\lambda = 0.3$/day, $R_E = R_0 = 3.87$. CI stands for confidence interval. **b** Probabilities for an infected individual being in each of the four phases on day $t$ after infection. The thick red line indicates the boundary between the pre-symptomatic and symptomatic phases. **c** Percentage of the infected population in each of the phases when the epidemic is growing at a rate $\lambda$. The thick red curve indicates the boundary between the pre-symptomatic and symptomatic population.

quarantined within a time window $t_{trace}$ since infection (Fig. 4b, blue line). This would bring the mean reproduction number $R_E$ from $R_0 = 3.87$ to a value below 1 if full tracing and quarantine is executed within 6 days after contact. An 80% tracing efficiency shrinks the time window to 3–4 days for achieving the same effect (Fig. 4b, red line). Details can be found in Supplementary Sections 4.1.2 and 4.1.3. The shaded areas on the plot, obtained from bootstrap analysis, show the range of the predicted reduction due to uncertainties in the incubation period estimation (see Supplementary Section 2.1.3).

**Social distancing and mask wearing**. Other than government-led interventions to break the transmission chain, individual-led efforts, including social-distancing, mask-wearing, frequent hand-washing, etc., can slow down or even stop the outbreak. Among them, radical shifts have taken place in people's attitudes towards population-wide mask wearing. It was practiced in most Asian countries since the initial phase of the outbreak, yet not adopted by the EU and USA until June 2020. As of August 2020, community mask use was recommended or required by most major public health bodies[25,26]. However, despite multiple experiments performed on measuring the trapping efficacy of masks on viral particles at individual's level[27–30] the aggregate impact of mask wearing at the population level is not yet clearly quantified. Given the now established risk of pre-symptomatic transmission, and the dominant role of droplet-mediated COVID-19 infections[31], masks with relatively low efficacy for personal protection may nevertheless reduce the overall infections in a population[32]. Based on a previous study on influenza aerosols[33], we constructed a semi-quantitative model to show that mask-wearing reduces $r(t)$ and hence $R_E$ by a factor $(1 - e \cdot p_m)^2$, where $e$ is the efficacy of trapping viral particles inside the mask, and $p_m$ is the percentage of the mask-wearing population (see Supplementary Section 4.2). According to this model, even for masks with intermediate efficacy ($e = 50\%$), population-wide mask-wearing at $p_m = 98\%$ alone could bring down $R_E$ from its basic value $R_0 = 3.87$ to 1, assuming no social segregation of mask-wearing and non-mask-wearing groups.

When combined with contact tracing (Fig. 4c), the two effects multiply. Figure 4c shows a heatmap of the reduced $R_E$ when contact tracing and isolation is completed within 5 days of infection. The solid black line indicates that the reduced $R_E$ reaches 1. For example, the combination of tracing of close contacts at 60% efficiency within 5 days and 60% of the general public wearing masks achieves the same purpose. This target line can be reached with lower percentages when close contacts can be found within 2 days of possible infection (dash-dotted line), but the numbers need to be higher when the time frame is relaxed to 8 days (dashed line).

**Provincial outbreaks and containment in China**. We examined the temporal progression of COVID-19 outbreaks in different

testing outcomes with one day delay (red curve). The largest reduction is obtained when the test is performed 3 days after the contact. This corresponds to the day when the width of the orange plus dark blue region in Fig. 3b is the widest.

For contact tracing and quarantine, we show our results under the scenario that a fraction $q_c$ of infectees are tracked down and

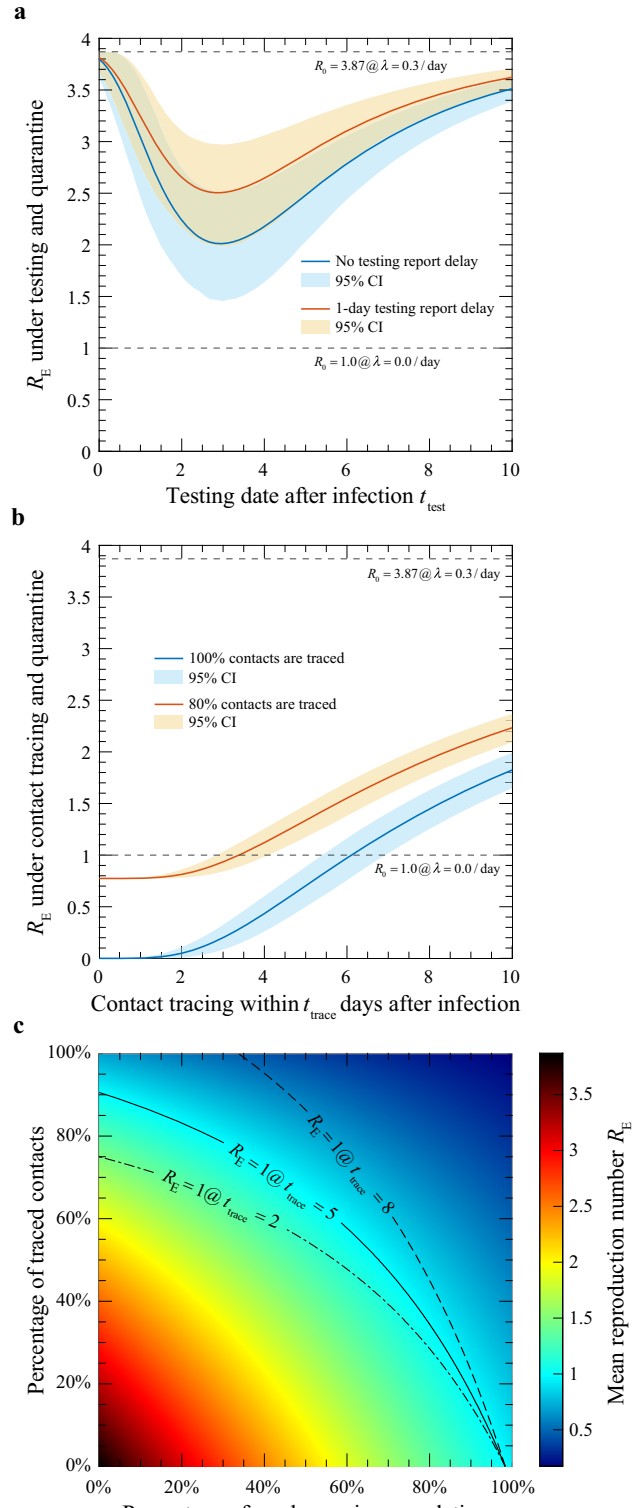

**Fig. 4 Reduction of the mean reproduction number upon intervention. a** Testing. Results are given for testing with 0 or 1 day reporting delay (blue and red curves), respectively. CI confidence interval. **b** Contact tracing and isolation. Results are shown for 100% (blue) and 80% (red) success rates, respectively. The 95% CIs of the estimated quantities in **a** and **b** (shaded areas) were obtained through bootstrap resampling with 1000 replications symptom onset times of 347 cases[11,20,21] and exposure windows of 66 transmission pairs[7]. **c** Mask-wearing in combination with contact tracing. The heatmap gives the reduced $R_E$ when contact tracing is implemented within 5 days after infection, assuming a basal value of 3.87. The solid black line marks the percentages required to reduce $R_E$ to 1. The dash-dotted line and the dashed line map out the percentages required to flatten the epidemic growth when the time frame for contact tracing is reduced to 2 days or relaxed to 8 days, respectively.

the epidemic as well as occurrences of smaller outbreaks. From the data, we define three phases of epidemic development.

Phase I is characterised by an exponential growth of the epidemic. In the first week after the Wuhan lockdown, nearly all provinces registered a growth rate of ~0.3/day (Fig. 5, region shaded in pink) in the newly confirmed cases. Reports indicate that most of the growth during this period was driven by imported cases from Hubei province, whose own growth continued at this rate for a longer period (Fig. 5a). The fraction of local infections during import-driven growth can be calculated and the result depends on the local value of $R_E$ through its mean reproduction rate Eq. (1) (see Supplementary Section 4.3).

Phase II is a crossover phase where public policies on border control and local intervention measures become increasingly stringent. On a logarithmic scale, data from the most affected provinces (except Hubei) show consistent behaviour. Closer examination, however, reveals the presence of sporadic outbreaks. Well-documented examples include prison cases in Hubei, Shandong and Zhejiang provinces[35]. Overall, under the swift and forceful implementation of COVID-19 surveillance, turn-around of the epidemic in provinces other than Hubei was reached in about 3 weeks after the Wuhan lockdown. In Fig. 5b and the Supplementary Fig. 5 (see Supplementary Section 4.4), we present simulation results using our model, assuming a linear decrease of $R_E$ from a local value of 2.0 to zero over a period $\Delta T$, which indeed reproduces the data in Fig. 5. The more gradual change of $R_E$ assumed in our simulations can be interpreted as due to the progressive mobility control and isolation policies including additional lockdowns, which took place from February 4 to 10, 2020[36,37], as well as allocation of massive resources by relevant authorities to conduct rigorous contact tracing and to rapidly expand isolation facilities for use by COVID-19 patients[38].

Phase III, or the final descent, occurred when the intervention measures essentially terminated transmission in the community. The few that re-emerged were quickly traced and contained. Within our model, the newly confirmed cases in this period are identified with the shrinking number of individuals moving from the latent to the symptomatic phase, as one moves along the time axis in Fig. 3b (see also Supplementary Section 4.5). Strikingly, the observed decay rate in this phase reached the maximum value of 0.31/day predicted by our model, including data from Hubei province shown in Fig. 5a. This observation indicates that the infected cases were isolated at extremely high efficiency. Interestingly, a similar decay in the daily new cases is seen on the cruise ship Diamond Princess (Fig. 5b).

**The first wave of COVID-19 outbreaks in other countries and regions.** Figure 6a–c show the daily confirmed cases in selected

parts of the world using the data available from the Johns Hopkins CSSE Repository[34], with the aim to extract more universal aspects of the pandemic development in light of our model studies. In the case of China, we focused on the daily confirmed cases from various provinces since the Wuhan lockdown on January 23, 2020. Broadly speaking, the ascending and descending curves follow very similar exponential laws, while the time it took to achieve the crossover was affected by the overall extent of

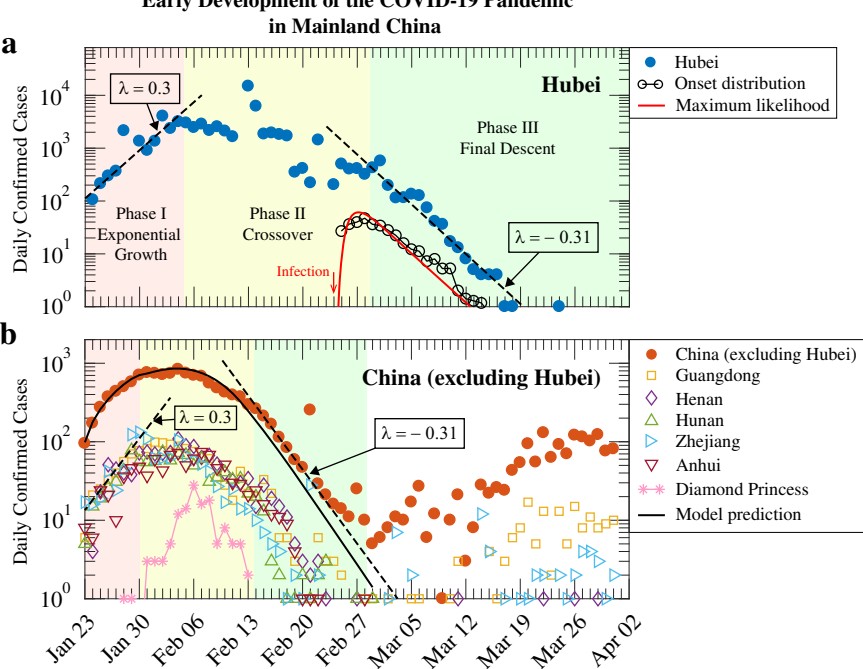

**Fig. 5 Growth and containment of the COVID-19 pandemic in mainland China.** Daily confirmed cases in Hubei and other provinces since the Wuhan lockdown on January 23, 2020. **a** Hubei province. The three phases of the epidemic development are shaded with different colours: exponential growth (red), crossover (yellow), and descent phase (green). Early exponential growth reached a rate $\lambda$ at ~0.3/day (left dashed line). Growth slowed and entered the crossover phase in the middle of the second week, and reached the third phase nearly 4 weeks later. The final descent that began in the beginning of March is characterised by $\lambda = -0.31$/day (right dashed line). The incubation period distribution is shown in open circles (reported data of 347 cases[11,21,22]) and red line (maximum-likelihood estimation) to compare with the exponential decay. Start of the incubation period is indicated by the red arrow. **b** Other provinces in China. The epidemic development in the affected provinces followed similar temporal patterns. Also shown is the model prediction of the daily confirmed cases (solid line), with details given in Supplementary Section 4.4. Newly confirmed cases from March onward (white region) are largely imported. Data for the Diamond Princess cruise ship[13] is included for comparison (asterisks).

countries and regions from late January till end of March 2020. Countries and regions in east Asia shown in Fig. 6a experienced the first wave sooner than the rest of the world, but the epidemic growth rate is much lower than other places due to the prevention measures in place such as border control and mask wearing by the general public. Despite these measures, South Korea documented a major outbreak in the second half of February that elevated the overall level of the epidemic in the country[39] (Fig. 6c). In countries in Europe and in the US, exponential growth of the pandemic, with a growth rate close to 0.3/day, were reported from the beginning of March onward (Fig. 6b), driven by local infections.

The surging pandemic triggered an emergency response by public health authorities and governments at all levels. Towards the end of March, countries that adopted stringent intervention measures have seen a significant reduction of the pandemic growth rate (Fig. 6b). The government of Italy imposed a national quarantine on March 9, 2020[40], after which growth in the number of newly confirmed cases slowed down[34]. On the other hand, South Korea implemented aggressive contact tracing and testing policies[41,42], enabling the country to bring the outbreak to a much-reduced level at $R_E \approx 1.0$.

In Fig. 6d we show the estimated epidemic growth rate $\lambda(T)$ against the cumulative number of confirmed cases $N(T)$ in five representative countries. We computed the growth rate from the local slope of the $\ln N(T)$ against $T$ curve, i.e., $\lambda(T) = \ln [N(T)/N(T-\Delta T)]/\Delta T$, using a time window $\Delta T = 3$ days. The interval between a few tens to a few thousands cumulative cases can be taken as the first phase of local outbreaks in these countries,

where the estimated values of $\lambda(T)$ remain approximately stable. Three of the five countries exhibited growth rates of ~0.3/day during this period, while Iran and Japan assumed values above 0.4/day and around 0.1/day, respectively. It is evident that epidemic preparedness and cultural aspects significantly affected COVID-19 spreading in the local population, before government intervention and containment measures took effect. A more complete discussion of growth rates during the exponential phase in different countries and regions can be found in Supplementary Section 5.

## Discussions

We have succeeded in developing a directly calibratable model for COVID-19 transmission by both pre-symptomatic and symptomatic viral carriers. This was made possible by focusing on transmission around the symptom onset, which is a prominent feature of the disease. Explicit mathematical expressions for the size of subpopulations in various phases of disease progression and the associated transmission risks are obtained. These results facilitate assessment of control measures, either to break the transmission chain or to reduce the overall level of social contacts in the community. For example, contact tracing, in combination with mask wearing in public places, can have a strong and immediate effect in bringing down epidemic growth. In reality, governments often take incremental steps in intervention measures to ease their impact on the economy and on people's livelihood. The quantitative treatment of epidemic control carried out in this study can serve as a reference in the decision-making process.

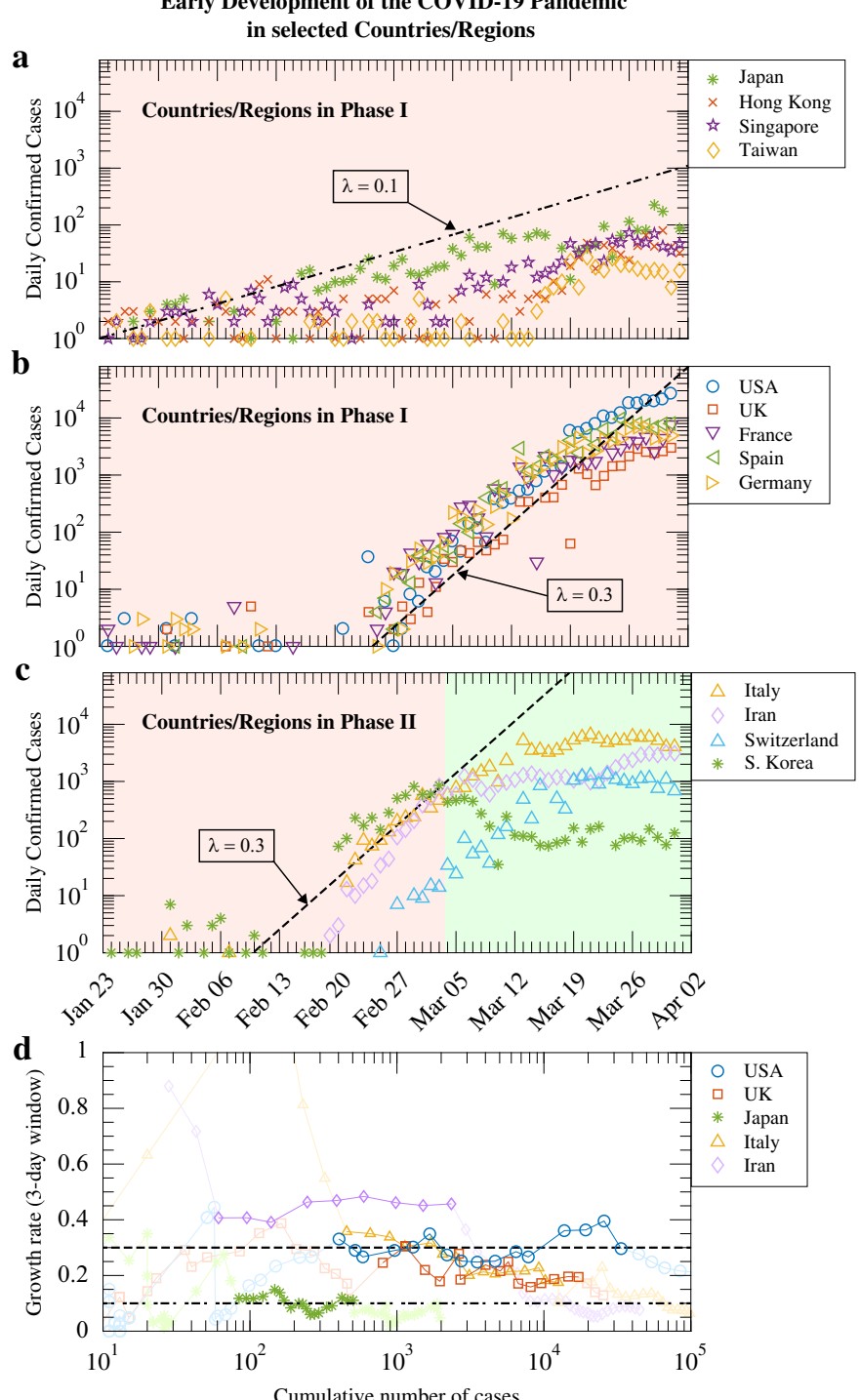

**Fig. 6 First wave of the COVID-19 pandemic in selected countries and regions. a–c** The number of daily confirmed cases from late January till end of March 2020. Countries/regions in **a** were successful in keeping transmission at a low level while those in **b** experienced exponential growth of local cases. Countries/regions in **c** have entered or been in the middle of phase II. Italy, South Korea, and Switzerland have reached zero or negative growth in daily confirmed cases, while data from Iran indicates a slowing down of the exponential growth. **d** The estimated epidemic growth rate $\lambda(T)$ against the cumulative number of confirmed cases $N(T)$ in five representative countries. Dashed and dashed-dotted lines indicate the exponential growth rates of 0.3/day and 0.1/day, respectively.

On a technical level, the modelling framework presented here is intuitive and flexible, and allows easy association of clinical features with population level pandemic development. This can be a significant advantage when the need arises to adapt the epidemic model to specific social environments and demographic composition. Our estimated incubation period distribution is in excellent agreement with other studies (see Table 1 for a comparison of key statistical features) and furthermore is not expected to change significantly over time. This places Eq. (1) as a convenient starting point for exploring temporal structures of epidemic development. The shift parameter $\theta_S$ in the equation embodies, in an explicit form, changing patterns of disease

transmission from symptomatic to the pre-symptomatic viral carriers, and hence can serve as an important index for epidemic control.

With regard to the quantitative predictions under specific intervention measures, the main uncertainty comes from estimation of their efficacy in reducing transmission from the infectious subpopulations identified in this study. As a baseline study, we estimated the infectiousness function $p_I(\Delta t)$ based on a relatively small data set of 66 transmission pairs which led to a sizable CI at 95% for its wings. This could improve as more carefully curated transmission cases during the initial outbreak become available. Response of the public to specific intervention measures is a complex topic that deserves extensive research in the future.

Finally, as with other epidemic models that assume a well-mixed population, our current modelling framework does not treat epidemic spreading in a heterogeneous population that exhibits complex spatio-temporal dynamics, nor does it consider significant differences in disease progression and transmission in different age groups. Some of the basic questions in COVID-19 epidemiological studies, such as whether pre-symptomatic spread constitutes a major contributor to disease transmission[43,44], cannot have definitive answers without considering these additional factors. In a large population, while individual outbreaks in specific communities may still follow the dynamics proposed here with suitable values of $R_E$, transmission across communities requires a separate treatment.

## Methods

**Key variables and parameters**. We collect key variables and parameters of the compartmentalised model together with the estimated values in Table 1 for easy reference.

**Incubation period distribution**. We analysed incubation periods of a total of $N = 347$ cases by combining three datasets[11,20,21]. For most cases, the infection date can only be assigned to a time window of more than one day. Therefore, the actual incubation period falls between $IPl_i$ and $IPu_i$, $i = 1, \ldots, N$, where $IPl_i$ and $IPu_i$ are the lower and upper bounds for case $i$. We perform maximum-likelihood estimation of the underlying symptom onset time distribution $p_O(t)$, following a scheme proposed by Reich et al. [47] Considering the exponential tail observed in the real data, we write

$$p_O(\theta, t) = \begin{cases} A p_{\text{left}}(t) & \text{for } t \leq t_e \\ A p_{\text{left}}(t_e) e^{-\gamma(t - t_e)} & \text{for } t \geq t_e \end{cases}, \quad (7)$$

where $A$ is the normalisation factor and $\theta$ denotes the set of parameters to be estimated. Transition to the exponential decay (with rate $\gamma$) takes place at $t_e$. Following common practice in the epidemiological literature, we take $p_{\text{left}}(t)$ to be a truncated log-normal or Weibull distribution with two parameters in each case

$$p_{\text{left}}(t) = \begin{cases} \frac{1}{t\sigma\sqrt{2\pi}} \exp\left[-\frac{(\ln t - \mu)^2}{2\sigma^2}\right] & \log-\text{normal} \\ \frac{k}{\lambda}\left(\frac{t}{\lambda}\right)^{k-1} \exp\left[-\left(\frac{t}{\lambda}\right)^k\right] & \text{Weibull} \end{cases}. \quad (8)$$

Continuity of derivatives at $t_e$ yields

$$\gamma = -\frac{p'_{\text{left}}(t_e)}{p_{\text{left}}(t_e)}. \quad (9)$$

Thus we are left with a set of three independent parameters. To estimate these parameters from the data, we consider the likelihood function

$$L(\theta; \mathbf{IP}) = \prod_{i=1}^{N} L(\theta; IPl_i, IPu_i), \quad (10)$$

with

$$L_i = L(\theta; IPl_i, IPu_i) = \int_{IPl_i - 0.5}^{IPu_i + 0.5} p_O(\theta, t) dt. \quad (11)$$

We performed optimisation and sensitivity analyses by scanning $t_e$ values from 4 to 8, and infinity for log-normal distribution and from 3 to 7, and infinity for Weibull distribution. The best estimate is obtained when $p_{\text{left}}(t)$ is a truncated log-normal distribution with $t_e = 6$ (see Supplementary Sec. 2.1 for details).

We also performed bootstrap analysis to determine uncertainties in the estimated $p_O(t)$. This is done by generating 1000 re-sampled copies of the original

dataset with 347 cases. The maximum-likelihood estimation of $p_O(t)$ is then performed for each of the re-sampled copies. The 95% CIs were obtained from the 1000 replications (see Table 1).

**Infectiousness profile**. Disease transmission is quantified by the infectiousness function $p_I(t)$, the probability density function for pairwise transmission at time $t$ since the symptom onset of the infector. We infer $p_I(t)$ by maximum-likelihood estimation, using the infector–infectee pairs published by He et al. [7]. In this dataset, the infectee exposure windows were documented in addition to the symptom onset dates of both infectors and infectees (77 pairs in total). Among them, 66 pairs have a unique symptom onset date (see Source Data), which are used here.

Given the general form and the limited temporal resolution of the dataset, we adopted simple exponentials for the two wings of the infectiousness function joined in the middle by a cap function,

$$p_I(\theta, t) = \begin{cases} A f(\theta_A) e^{\alpha_A(t - \theta_A)} & t \leq \theta_A \\ A f(t) & \theta_A \leq t \leq \theta_B \\ A f(\theta_B) e^{-\alpha_B(t - \theta_B)} & t \geq \theta_B \end{cases}, \quad (12)$$

where $A$ is the normalisation factor. The infectiousness function transits to the left exponential tail at $\theta_A$ and to the right exponential tail at $\theta_B$. Between $\theta_A$ and $\theta_B$, it takes the form of $f(t)$. We consider three different forms of $f(t)$:

- Model 1: $f(t) = 1$ and $\theta_A = \theta_B = \theta_P$ (two exponential tails directly join at $\theta_P$); Independent parameters $\theta = (\alpha_A, \alpha_B, \theta_P)$.
- Model 2: $f(t) = 1$ and $\theta_A < \theta_B$ (two exponential tails with a flat cap of length $\epsilon = \theta_B - \theta_A$, centred at $t_P$); Independent parameters $\theta = (\alpha_A, \alpha_B, \epsilon, \theta_P)$.
- Model 3: $f(t) = [1 - \chi(t - \theta_P)^2]$ and $\theta_A < \theta_B$ (two exponential tails with a rounded cap peaked at $t\theta_P$, whose shape is characterised by $\chi$); Independent parameters $\theta = (\alpha_A, \alpha_B, \chi, \theta_P)$ ($\theta_A$ and $\theta_B$ are determined by the smoothness condition).

We perform maximum-likelihood estimations using the dataset mentioned above, where each transmission pair $i$ is associated with an exposure window $W_i = [Wl_i, Wu_i]$ relative to the symptom onset of the infector. The likelihood function is constructed as follows:

$$L(\theta; \mathbf{W}) = \prod_{i=1}^{N} L(\theta; Wl_i, Wu_i), \quad (13)$$

where

$$L_i = L(\theta; Wl_i, Wu_i) = \int_{Wl_i - 0.5}^{Wu_i + 0.5} p_I(\theta, t) dt. \quad (14)$$

Sensitivity analysis is performed at a set of values for $\epsilon$ (Model 2) and $\chi$ (Model 3), respectively. In both cases, the best estimate degenerates into Model 1 (see Supplementary Section 2.2 for details).

The uncertainty in the estimated $p_I(t)$ is determined through bootstrapping with 1000 replications, with which the 95% CIs were obtained (see Table 1).

**Reporting summary**. Further information on research design is available in the Nature Research Reporting Summary linked to this article.

## Data availability

The COVID-19 pandemic and clinical data used in this work are all from published studies or datasets: confirmed case numbers in selected countries/regions (COVID-19 Data Repository by the Center for Systems Science and Engineering at Johns Hopkins University: https://github.com/CSSEGISandData/COVID-19), confirmed cases number for the Diamond Princess cruise ship[13], incubation period (three datasets[11,20,21]), exposure window of transmission (one dataset[7]), and serial interval (one dataset[22,23]). These datasets are included in Source Data for easy reference, together with the parametrised distributions and the $R_E - \lambda$ relation from this work.

## Code availability

MATLAB codes for the analyses presented are deposited at: https://github.com/hkbu-covid19group/Calib-covid19[48].

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

## Acknowledgements

The work is supported in part by the NSFC under Grant Nos. 11635002, 32000886, and U1930402, and by the Research Grants Council of the Hong Kong Special Administrative Region (HKSAR) under Grants HKBU 12324716 and 12304020. Any views expressed by Jiang Liu and Viola Tang are not as a representative speaking for or on behalf of his/her employer, nor do they represent his/her employer's positions, strategies or opinions.

## Author contributions
L.-H.T., X.F.L. and J.L. formulated and analysed the mathematical model. L.T. performed MLE and other statistical analyses of the data. Z.L. constructed and analysed the face-mask model. L.-H.T., L.T., X.F.L., V.T., Z.L., H.L., F.Q., and J.L. wrote the paper. L.-H.T. coordinated the project. All authors participated in the pandemic and clinical data collection and initial analysis, and the discussion of results.

## Competing interests
The authors declare no competing interests.
