## [Peer Review File · Nature Communications]

Reviewers' Comments:

Reviewer #1:

Remarks to the Author:

General comments:

This is an analytical study of characteristics of transmission and how those might impact control of COVID epidemics.

The paper is interesting but there is insufficient quantification of uncertainty in any of the parameters. And the authors do not present sensitivity analyses on some of the assumptions, e.g. selected distributions.

The treatment of some the interventions mentioned is too cursory, with implicit assumptions in the supplementary information, and these assumptions are not discussed as limitations, i.e. possible overestimation of effect of masks. There are very few limitations discussed of the framework or assumptions.

Specific comments:

Suggest to add a table of parameter names with interpretation for the model.

Suggest to split the model section up.

Suggest Figure 1 could be the other way up. The legend talks about transmission, then disease, then transmission again, then intervention, but the figure is out of order.

<https://www.nature.com/articles/s41591-020-0869-5> This paper may also be of interest for fitting the method.

How sensitive are the R_0 estimates to other parameters of the model? Can sensitivity analyses be shown? Can some estimate of uncertainty (i.e. confidence interval) be given for the R_0 estimate of 3.68?

"The data is validated against a serial interval study on 468 infection pairs 9 with excellent consistency (see SI)." This should be in the main text. As should more explanation/justification of if this is exponential decay or if any other functions fit as well. S2 and S3 (model fits) should be in the main text. And some quantification of uncertainty.

How sensitive is the optimal day to the shape of the distribution of R_A and R_S ? How much uncertainty is there over the course of 1 day (e.g. 24h), given that the timelines are quite short in COVID and especially around the durations that the model is focussing on?

Suggest to split up the evaluation of intervention measures section, because there are a lot of interventions covered here.

Fig 3b would be much more useful if the contact tracing line had uncertainty on it, e.g. from parameter uncertainty, either from fitted or simulated parameters.

Masks: this is perfect utilisation of masks and doesn't account for errors in wearing them, e.g. touching outside, reusing, etc, that could affect efficacy. This analysis is too optimistic in the effect of masks. Where is the evidence for protection of the uninfected from masks?

Fig 4 and Fig S5. Bell shaped curve is usually used for Normal distributions, which are symmetric.

These don't look particularly symmetric.

Fig 4c, suggest fitting the exponential phase to countries individually, and potentially regionally (as done in China in 4b).

"Rampaging" is unnecessary.

Supplement 1.5. "Wallinga" not Wallinger.

Reviewer #2:

Remarks to the Author:

The authors develop a novel model which has some reasonable properties, but no obvious advantages over existing models. The major innovations are:

* the approximation (1), which seems poorly motivated (and the authors stress that it's not actually important and is relaxed in the supp)

* the strange and poorly discussed definition of β_{eff} , which is calculated by assuming R_S can be calculated, then multiplying that quantity by α_A , apparently for the purpose of being able to divide by α_A after adding. (2)

Otherwise, the authors are just doing a standard renewal-equation model with ad hoc parameters (many based on one or two papers, sometimes not published) and no propagation of uncertainty.

The advantages of the their approach in terms of physical interpretation of parameters is not "obvious" to me and is not clearly discussed.

The "Evaluation" section involves applying naive assumptions about control to a point estimate of R_0 , with textbook results. The advantages of mediating this argument through β_{eff} are not clear.

The authors point that a slope of $-0.32/\text{day}$ is exactly matched by two time series is not very convincing without statistics; the observation that this would be expected under the authors' assumptions when R is equal (or extremely close to) 0 should not be used as evidence for those assumptions in the absence other evidence that R was extremely close to 0.

Jonathan Dushoff

Reviewer #3:

Remarks to the Author:

The manuscript concerns an interesting model dividing the time from infection to recovery and death into different parts: latent, infectious no symptoms and infectious with symptoms, and estimates the amount of infection from the latter two and what effect different preventive measures may have.

Major comments

It is a massive simplification to think that decay rate -0.32 is "universal, and also that most countries have an increase of $\lambda=0.30$. This must be weakened. Many countries did not see much rise at all even before preventive measures. My impression is that $\lambda=0.3$ applies to countries region with highest spreading. Similarly, I see no reason why $R_0=3.68$ should apply to

all countries.

I did not understand how β_A and β_S were estimated

One problem is that reported number of cases is not the same as number of infected. This is of course always true but should be acknowledged better. For instance, I am quite convinced that the severe lock-down in Hubei actually did drop infections with a discrete big jump down. But the same will of course not be true for reported cases, since individuals carry the virus for about 5-7 (?) days.

You estimate the effect of wearing face masks alone and in combination of other preventive measures. For this you have a quantity e describing its efficacy in terms of protection. I could not find any numerical estimate of it in the report, not citations in the literature to support your choice.

Minor comments

Update case and death figures and stress that the former is confirmed cases and that the true number is magnitudes higher.

Coming from mathematics "bell shaped curve" is for me reserved to the Gaussian distribution which has nothing to do with the current curve. Of course, the current curve also resembles that of a bell ...

What is motivation for having time varying rate to end latent period but not for A-state?

Does β_{eff} have an interpretation?

Just before discussion section: I agree with Korea might having import driven growth but not regarding Italy. Please argue why you think this is the case (or remove Italy)

Fig 3, caption: "... to flatten epidemic growth". Please clarify by saying that the reproduction is reduced to below 1.

Author Replies to Reviewers' Comments

We thank all three reviewers for their careful reading of the manuscript and for their insightful and constructive comments and suggestions, which helped us tremendously in improving various aspects of the manuscript. Before addressing the specific comments, we summarise here our key responses:

- A suggestion from all three reviewers is to include an uncertainty analysis of key model parameters. This is indeed an issue of great importance which was not adequately addressed in our previous submission. In the revised manuscript, we re-examined the raw data in cited references in combination with datasets that have since appeared. Maximum likelihood estimations of model parameters were performed. The associated confidence intervals were determined using the standard bootstrap method. The results are collected in a new Table I. When appropriate, we compared our results with those in published work and found good consistency.
- We have clarified and strengthened the connection between the simple and intuitive COVID-19 transmission model defined by Eq. (1) and the compartmentalised model that allows quantification of subpopulations for epidemic control. A new parameter θ_S is introduced in Eq. (1) to accommodate shifts in transmission from largely symptomatic in the early days to increasingly pre-symptomatic. Epidemic predictions can be adjusted following the recalibration steps described in the paper.
- Regarding the effects of mask-wearing, we include more recently published papers to further support our analysis. We have considered additional factors that could modify the effect of population-wide mask-wearing, such as infections relying on nonrespiratory routes and the different filtering efficacies for masks as source control and personal protection. More details are presented on the following website:
http://www.zhiyuanlab.xyz/MASK_0906.html .

We would like to point out that although a large number of COVID-19 papers have appeared since our first submission, the basic premise of our paper still stands. We were more optimistic that better data would lead to more accurate values for the model parameters, but our study in the past four months led to the realisation that the actual values of these parameters are affected by many factors such as age, health and living conditions, as well as initial viral dosage, etc. In revising the manuscript, we have included uncertainty estimates wherever possible, but would also like to encourage more systematic studies of demographic and environmental factors so as to significantly improve the accuracy of our model predictions.

Regarding the reviewers' concern in “overestimation of the effect of masks”:

First, we considered the situation where the filtering efficacy of masks are relatively low, and asked whether masks with intermediate efficacy in personal protection (for example, only filtering ~50% of virus particles) can have an aggregate effect when applied on a population-wide scale. In Fig S3, impacts of masks with different filtering efficacy are considered. The results show that even masks that only trap 20% of the virus particles can yield a significant impact when widely adopted by the population. Also, our previous mask model is highly simplified in order to deliver the essential message. In the model shown in http://www.zhiyuanlab.xyz/MASK_0906.html, more details related to the effect of population-wide mask-wearing were considered, such as infections relying on nonrespiratory routes, and the different filtering efficacies for masks as source control and personal protection.

There are already several experimental measurements on the efficacy of different types of masks against coronavirus, both as source control (SI-Ref. 15,17,18) and personal protection equipment (SI-Ref. 16,19,20). In summary, there is laboratory-based evidence that surgical or N95 masks have satisfying filtration capacity in the relevant droplet size range of coronavirus. The experimental reports are included in the references.

Several recent modeling works focusing on the effect of population-wide mask-wearing converge to similar conclusions that masks of intermediate filtering efficacy exhibit aggregate effect at the population level. For example, in the work of Stutt *et al.* (SI-Ref. 21), they found that with a policy that all individuals must wear masks at all times, a median effective COVID-19 R_0 of below 1 could be reached, even with mask effectiveness of 50% (for $R_0=2.2$) or mask effectiveness of 75% (for $R_0=4$). Similarly, models from Kai *et al.* (SI-Ref. 22) estimated that 80-90% masking would eventually eliminate the disease. Work from Fisman *et al.* (SI-Ref. 23) also showed similar results.

Response to Reviewer #1

General comments:

This is an analytical study of characteristics of transmission and how those might impact control of COVID epidemics. The paper is interesting but there is insufficient quantification of uncertainty in any of the parameters. And the authors do not present sensitivity analyses on some of the assumptions, e.g. selected distributions.

Response: We thank the reviewer for appreciating the contribution of our work towards quantification of COVID-19 transmission and targeted control measures. Both the degree of uncertainty in the estimated model parameters and the sensitivity of our results against such uncertainties are indeed important issues. We have taken time to re-examine the model calibration against the previously employed and more recent data. In the revised manuscript, we present maximum likelihood estimations of the symptom onset time distribution (Fig. 2a) and the infectiousness function (Fig. 2b) using raw data in published literature. We have also revised our compartmentalised model (Fig. 1b and SI) to better accommodate the empirical observations. Estimated values of model parameters, including their confidence intervals, can be found in Table I, Methods section. The effect of the sample-related uncertainties on our model predictions, i.e., the R_E versus λ curve (Fig. 3a) and the effective mean reproduction number under respective intervention measures (Figs. 4a and 4b), are now presented.

The treatment of some of the interventions mentioned is too cursory, with implicit assumptions in the supplementary information, and these assumptions are not discussed as limitations, i.e. possible overestimation of effect of masks. There are very few limitations discussed of the framework or assumptions.

Response: We thank the reviewer for raising this issue. Impacts of various types of interventions on COVID-19 transmission were not quantitatively available while the initial manuscript was prepared. Therefore we focused on the derivation of general expressions which can be applied to specific situations when the relevant data are known. Selected outcomes of these expressions were illustrated for easy reference. As more in-depth research has been carried out and published, the assumptions have generally been validated. In the revised manuscript, we updated the relevant sections and added new references. Limitations of our approach are also mentioned where appropriate.

The reviewer is correct that the efficacy of face masks depends on numerous factors. Exactly how much face masks contributed to the slowing down of the pandemic in countries where mask-wearing is prevalent is debatable. As Reviewer #3 had a similar enquiry, we presented our current knowledge about mask-wearing in the summary response above.

Specific comments:

1. Suggest to add a table of parameter names with interpretation for the model.

Response: Very good suggestion. The information is now included in Table I of the Main Text.

2. Suggest to split the model section up.

Response: Very good suggestion. We have rewritten the model section in the Main Text to better explain our rationale and strategy, which has been structured into two parts, model description and parameter calibration.

3. Suggest Figure 1 could be the other way up. The legend talks about transmission, then disease, then transmission again, then intervention, but the figure is out of order.

Response: As suggested, we have rearranged Fig. 1 to better align with the text.

4. <https://www.nature.com/articles/s41591-020-0869-5> This paper may also be of interest for fitting the method.

Response: Suggestion appreciated. The medRxiv version of this work was cited in the original submission. In the revised manuscript, the transmission pairs reported by He *et al.* were the primary source we used to obtain the maximum likelihood estimation of the infectiousness function (Fig. 2b). Consistency of the result obtained with a much larger dataset on serial interval is seen in Fig. 2c.

5. How sensitive are the R_0 estimates to other parameters of the model? Can sensitivity analyses be shown? Can some estimate of uncertainty (i.e. confidence interval) be given for the R_0 estimate of 3.68?

Response: Within our modeling framework, R_0 is given by Eq. (5) of the updated Main Text. Thus uncertainties in R_0 are inherited from uncertainties in the incubation period distribution and from that of the shift parameter θ_S . The latter is given approximately by Eq. (4), with quantities on the right-hand-side extracted from the shape of the infectiousness function. Using a bootstrapping method, we estimated uncertainties in the incubation period distribution (Fig. 2a). The estimated value of R_0 is 3.87 at a growth rate of 0.3/day, with the 95% confidence interval from 3.38 to 4.48.

6. “The data is validated against a serial interval study on 468 infection pairs with excellent consistency (see SI).” This should be in the Main Text. As should more explanation/justification of if this is exponential decay or if any other functions fit as well. S2 and S3 (model fits) should be in the Main Text. And some quantification of uncertainty.

Response: We thank the reviewer for the suggestion. We have added a new figure in the Main Text to reflect this point (Fig. 2c in the updated manuscript). The figure is accompanied by a new subsection named “Serial interval”.

7. How sensitive is the optimal day to the shape of the distribution of R_A and R_S ? How much uncertainty is there over the course of 1 day (e.g. 24h), given that the timelines are quite short in COVID and especially around the durations that the model is focussing on?

Response: Inspired by this comment, we revisited the raw pairwise transmission data collected by He *et al.* and carried out a detailed maximum likelihood estimation of the infectiousness function. Peak infection was found to be a little less than 1 day prior to the symptom onset (SI Sec. 2.2). The quality and quantity of the data are not sufficient to yield a definitive infectiousness function, which itself can vary over time as reported in Ref. 7. We parameterised the two wings of the curve with exponentially decaying functions. Estimated values of the parameters and their CI are given in Table I. Larger and better-quality datasets are certainly desirable, though there could be intrinsic limits as the reviewer noted.

8. Suggest to split up the evaluation of intervention measures section, because there are a lot of interventions covered here.

Response: Following the reviewer’s suggestion, we have updated the section on the evaluation of intervention measures, and re-organised the material into two subsections, (1) testing and contact tracing, and (2) social distancing and mask wearing.

9. Fig 3b would be much more useful if the contact tracing line had uncertainty on it, e.g. from parameter uncertainty, either from fitted or simulated parameters.

Response: The original Fig. 3b is now Figs. 4a and 4b. Based on our analytical results (Eq. S37), the uncertainties mainly come from two parts: 1) θ_S , the time shift of the infectiousness $r(t)$ with respect to the original incubation period distribution, which is determined by the infectiousness profile around the symptom onset, and 2) the distribution of the incubation period. We have included the uncertainty in the updated figures (Figs. 4a and 4b, shaded areas).

10. Masks: this is perfect utilisation of masks and doesn't account for errors in wearing them, e.g. touching outside, reusing, etc, that could affect efficacy. This analysis is too optimistic in the effect of masks. Where is the evidence for protection of the uninfected from masks?

Response: In our analysis we did not assume perfect utilisation of masks. Actually, we considered the situation where one mask could only block 20%~50% of the virus, which is much lower than the standard surgical mask when appropriately used [Ref. 29 in the Main Text]. We are interested in whether masks with unsatisfying efficacy in personal protection can have an aggregated effect when applied on a population-wide scale. In Fig S3, masks with different filtering efficacy are considered, which shows that even masks which trap 20% of virus particles could show a significant effect when generally adopted by the population.

Additionally, there are two layers of effects for masks: source control and personal protection. The first layer reduces viral shedding from potentially infectious people, which is critical given the significant proportion of pre-symptomatic transmission of COVID19. Experimentally, there are already several measurements on the efficacy of different types of masks against coronavirus, both as source control (Ref. 27, 28 in the Main Text and SI-Ref. 18) and as personal protection equipment (Ref. 29, 30 in the Main Text and SI-Ref. 20). The ability of masks to filter particles depends on the particle size and trajectory, with smaller floating aerosols being more challenging to filter than larger ones. In the laboratory setting, there is evidence that masks can filter in the relevant droplet size range for COVID-19, as well as efficacy in blocking droplets and particles from the wearer (Ref. 29 in the Main Text and SI-Ref. 13, 14). Personal protection is more challenging than source control, since the inhaling particles are smaller. According to the World Health Organization's "Advice on the use of masks in the context of COVID-19" (Ref. 30), the penetration for surgical masks is 50%-60%, which is the range we used in our simplified model.

11. Fig 4 and Fig S5. Bell shaped curve is usually used for Normal distributions, which are symmetric. These don't look particularly symmetric.

Response: Thanks for this remark. We have removed the expression of the bell shaped curve from the manuscript and the SI.

12. Fig 4c, suggest fitting the exponential phase to countries individually, and potentially regionally (as done in China in 4b).

Response: We have reorganised the original Fig. 4c into two parts in the revised manuscript (Fig. 5c). Furthermore, the cumulative case numbers instead of the daily new cases are shown to allow better comparison with exponential growth after an outbreak has reached more than 100 in the number of confirmed cases. With mask-wearing widely practiced, the early epidemic growth rates of Asian countries/regions were much lower.

13. “Rampaging” is unnecessary.

Response: We have revised the wording.

14. Supplement 1.5. “Wallinga” not Wallinger.

Response: Thanks for pointing out the typo which has been corrected.

Response to Reviewer #2

General comments:

The authors develop a novel model which has some reasonable properties, but no obvious advantages over existing models. The major innovations are:

* the approximation (1), which seems poorly motivated (and the authors stress that it's not actually important and is relaxed in the supp)

* the strange and poorly discussed definition of β_{eff} , which is calculated by assuming R_S can be calculated, then multiplying that quantity by α_A , apparently for the purpose of being able to divide by α_A after adding. (2).

Otherwise, the authors are just doing a standard renewal-equation model with ad hoc parameters (many based on one or two papers, sometimes not published) and no propagation of uncertainty.

Response: We thank the reviewer for his careful reading of our paper.

The whole experience with COVID-19 is a learning process that is likely to continue for quite some time into the future. As is well known by now, the epidemiological characteristics of COVID-19 change over time and also are shaped by the local policies and way of life. Therefore, a model that is simple and intuitive, solvable, and more importantly, integrates clinic observations for continuous update of parameters, has a clear advantage over those that rely on epidemiological time series analysis alone.

The model presented in this work is one of a few that lay a direct pipeline from clinical data to epidemic development and onto control measures. Not only is this a novel approach, but also renders significant advantages over existing models by enabling policymakers to dynamically calibrate containment measures over time. In the resubmitted manuscript, we have substantially revised the model presentation to make the above procedure more transparent and easier to follow.

The major innovation of our paper is demonstrated by Eq. (1), which embodies the quantitative characteristics of COVID-19 transmission, i.e., the incubation period distribution, the shape of the infectiousness curve, and the mean reproduction number in a single expression. The parameter θ_S , which was buried in the calculation in SI before, is now given explicitly in Eq. (1). This equation is derived from the more detailed model that matches clinical observations. Uncertainty analysis is now included for the key model parameters and many of our predictions.

The parameter β_{eff} was previously introduced to play the dual role of setting the overall scale of disease transmission as well as lumping the pre-symptomatic and post-symptomatic transmissions into transmission at the symptom onset. This is indeed a bit confusing to the reader without going through the technical discussions in the SI. In the revised manuscript, we have considerably expanded the model description and the parameter calibration, now presented in two separate sections.

A new Fig. 1a is added to illustrate our rationale in model construction. With these improvements, we believe the novelty and uniqueness of our work are made clear.

Specific comments:

The advantages of their approach in terms of physical interpretation of parameters is not "obvious" to me and is not clearly discussed.

Response: This comment is partially addressed above. For easy reference, we have collected parameter definition, estimated values and their CI in Table I in Methods. We have significantly expanded the description of parameter estimation in the Methods section and in the SI.

The "Evaluation" section involves applying naive assumptions about control to a point estimate of R_0 , with textbook results. The advantages of mediating this argument through β_{eff} are not clear.

Response: The purpose of our paper is not to exhaustively review the infinite combination of containment measures and factors that affect COVID-19 epidemic development for every locale at every point in time. Rather, we present a novel modelling approach and illustrate in the Evaluation section how the model can be applied to common containment measures under a specific example situation, namely when $R_0 = 3.87$.

We took the situation where $R_0 = 3.87$ (95% CI [3.38, 4.48]), derived from our model for when the epidemic is at an exponential growth rate of 0.3/day, because this growth rate was seen in the early epidemic data from a number of countries shown in Fig. 5c. Other values of R_0 can be used in our model to evaluate the impact of different containment measures on stemming epidemic development at various stages and locations.

The authors point that a slope of -0.32/day is exactly matched by two time series is not very convincing without statistics; the observation that this would be expected under the authors'

assumptions when R is equal (or extremely close to) 0 should not be used as evidence for those assumptions in the absence other evidence that R was extremely close to 0.

Response:

We agree with the reviewer that the statement can be softened to accommodate for a weak residual transmission. In the revised manuscript, uncertainty analysis of the model parameters and their propagation in the modeling results were carried out. The revised rate at zero transmission from our model is now -0.31/day (95% CI [-0.35, -0.27]). Although extremely low transmission rate towards the end of the first wave in China is likely, some cautious remarks are included for a factual interpretation of our findings.

Response to Reviewer #3

The manuscript concerns an interesting model dividing the time from infection to recovery and death into different parts: latent, infectious no symptoms and infectious with symptoms, and estimates the amount of infection from the latter two and what effect different preventive measures may have.

We thank Reviewer #3 for reviewing our manuscript and for her/his positive assessment on our work. We next address each of her/his concerns in order.

General comments:

1. It is a massive simplification to think that decay rate -0.32 is "universal, and also that most countries have an increase of $\lambda=0.30$. This must be weakened. Many countries did not see much rise at all even before preventive measures. My impression is that $\lambda=0.3$ applies to countries region with highest spreading. Similarly, I see no reason why $R_0=3.68$ should apply to all countries.

Response: We thank Reviewer #3 for this constructive comment. We have revised the discussion and weakened the statements as suggested (please see Main Text Pages 7 and 8).

The decay rate $-0.32/\text{day}$ at the end of the outbreak is based on the scenario that all transmissions are stopped after a certain date. Based on a detailed uncertainty analysis presented in the revised manuscript, the predicted decay rate under this scenario is now revised to $-0.31/\text{day}$ (95% CI $-0.35/\text{day}$ to $-0.27/\text{day}$). The observed decay rate in China and in Hubei province supports such a scenario but this is certainly not the only way to end the COVID-19 outbreak. The last point should be clear to the reader.

With regard to the epidemic growth rate at the very early stage of COVID-19 outbreak in a given region, we agree with the reviewer that there are many factors that can influence the result. Countries that had recent experience with outbreaks of this type, notably SARS and MERS, were better prepared in this period. It is widely believed that such awareness has contributed to the behaviour shown in Fig. 5c, right panel. On the other hand, in populous regions that did not adopt major precautionary measures at this stage, the growth rate did reach around $0.3/\text{day}$. With improved calibration of the model parameters, the basic reproduction number R_0 is estimated to be around 3.87 (95% CI 3.38 to 4.48) for these countries/regions.

2. I did not understand how β_A and β_S were estimated

Response: We thank Reviewer #3 for the question. The two parameters were introduced to quantify the level of pre-symptomatic and post-symptomatic transmission in our compartmentalised model. As far as we know, there is not yet published data that allows us to estimate them directly. Instead, they could be estimated indirectly from the infectiousness function and the mean reproduction number. The former can be estimated from case studies, while the latter is a variable we use to relate back to the rate of epidemic growth.

We have substantially revised and improved the manuscript to explain the modelling and parameter calibration details. All the key variables and parameters are collected in Table I in the Method section for easy reference.

3. One problem is that reported number of cases is not the same as number of infected. This is of course always true but should be acknowledged better. For instance, I am quite convinced that the severe lock-down in Hubei actually did drop infections with a discrete big jump down. But the same will of course not be true for reported cases, since individuals carry the virus for about 5-7 (?) days.

Response: We fully agree with Reviewer #3 that the reported/confirmed number of cases is not the same as the number of infected cases. In the exponential growth phase, one may argue that the two are proportional to each other when testing and intervention measures are implemented at certain efficiency. More dramatic measures such as lockdown can change the course of an outbreak substantially, as we illustrated in Sec. 4.7 and Fig. S5 of the SI. During revision, we have tried to avoid possible mis-understanding or overstatement of our findings.

4. You estimate the effect of wearing face masks alone and in combination of other preventive measures. For this you have a quantity e describing its efficacy in terms of protection. I could not find any numerical estimate of it in the report, not citations in the literature to support your choice.

Response: The estimation of e is estimated based on information provided by Ref. 27, 30 in the Main Text as well as SI-Ref. 13, 14. Additional references are cited in the Main Text and SI for more evidence.

There are two layers of effects for masks: source control and personal protection. The first layer reduces viral shedding from potentially infectious people, which is critical given the significant proportion of pre-symptomatic transmission of COVID19. In our very simplified mathematical model, the efficacy of masks in source control and personal protection, represented as the percentage of virus particles blocked by masks, were set to be equal (e is the symbol for efficacy). More complete models considering different efficacy in blocking inhaling and exhaling virus and nonrespiratory transmission can be considered, but it is not the main focus of

this work. In this work, we are interested in if imperfect mask-wearing (only 50% of the virus blocked) can have an aggregated effect when adopted on a population-wide scale. The ability of masks to filter particles depends on the particle size and trajectory, with smaller floating aerosols more challenging to filter than larger ones. In the laboratory setting, there is evidence that masks are able to filter in the relevant droplet size range for COVID19, as well as efficacy in blocking droplets and particles from the wearer in a range higher or near the efficacy of 50%. (SI-Ref. 13, 14). For seasonal coronaviruses, surgical masks for source control were effective at blocking coronavirus droplets of all sizes for every subject (Ref. 27 in the Main Text).

Personal protection is more challenging than source control, since the inhaling particles are smaller. According to World Health Organization's "Advice on the use of masks in the context of COVID-19" (Ref. 30 in the Main Text), the penetration for surgical masks is 50%-60% , which is the range we used in our simplified model.

Specific comments:

1. Update case and death figures and stress that the former is confirmed cases and that the true number is magnitudes higher.

Response: Following the reviewer's suggestion, we have revised the wording to reflect this and updated the figure captions where required.

2. Coming from mathematics "bell shaped curve" is for me reserved to the Gaussian distribution which has nothing to do with the current curve. Of course, the current curve also resembles that of a bell ...

Response: We thank Reviewer #3 for the comment. The wording is changed.

3. What is motivation for having time varying rate to end latent period but not for A-state?

Response: This is an excellent question. The main assumption in our model construction is based on the clinical observation that viral transmission mostly takes place within a few days before or after the symptom onset. Data on pre-symptomatic transmission are still quite incomplete and their interpretations contested (see Refs 7, 41, 42). However, using the symptom onset instead of the infection time as the anchor point of a patient's viral transmissibility appears to be in agreement with the majority of clinical evidence.

To build a stochastic model of transmission at the individual level, we adopted the strategy of restricting pre-symptomatic transmission to an interval of variable length from the symptom

onset point. A constant transmission rate, coupled with the Markovian dynamics for the length of this infectious period, yields a mean infectiousness curve that decays exponentially on the pre-symptomatic side. In the revised manuscript, we introduced an additional compartment A_2 to account for the fact that the peak position of the infectiousness curve may not overlap with the symptom onset. This extension allows our model to make more quantitative predictions and also to be able to adapt to changing transmission patterns as post-symptomatic transmission is significantly reduced.

Based on our understanding of the pathophysiology of COVID-19, viral infection and disease progression goes through a number of intermediate stages whose durations vary greatly from person to person. So a non-Markovian model for the whole incubation period is preferred. Given our definition of the A phase, the non-Markovian nature is shifted to the latent period. We note that this is more of a matter of convenience (avoiding unnecessary and un-substantive complications) given the quality of data at hand and the level of understanding at present.

4. Does β_{eff} have an interpretation?

Response: With the introduction of θ_S and derivation of Eq. (1) from the compartmentalised model, there is no need to keep β_{eff} which was originally introduced for technical reasons.

5. Just before discussion section: I agree with Korea might having import driven growth but not regarding Italy. Please argue why you think this is the case (or remove Italy)

Response: We thank Reviewer #3 for pointing this out. We have revised “These countries” to “In some instances, this behaviour could be attributed to imported cases as mentioned above.”

6. Fig 3, caption: "... to flatten epidemic growth". Please clarify by saying that the production is reduced to below 1.

Response: Thank you for pointing this out. We have updated the caption as suggested (Fig. 5 in the revised manuscript).

Reviewers' Comments:

Reviewer #2:

Remarks to the Author:

I continue to struggle with this paper.

The foundational part (3) seems just wrong to me. The authors frame their model as a renewal equation with kernel K , and then show in the supp that K can be calculated if $r(t)$ is known. This seems practically wrong: in particular, if there is a regime where $r(t)$ is constant, that should be consistent with a broad range of kernel functions, but (3) seems to show that a constant $r(t)$ produces a constant kernel! There is also a weird conflation, since time in $r(t)$ is calendar time, but the primary t in the kernel function is a delay time; these values should not be directly comparable without an offset.

S5 seems just wrong, since the last line seems to show that $r(t)$ is constitutively positive (I guess it is missing a term of α). I think this problem is fixed by S7, but it cost me a bunch of time.

I guess S7 is right in some universe of assumptions not clearly laid out by the authors. They are showing that there is some world where $K(t)$ can be inferred from $r(t)$. As mentioned above, this is an unnatural thing to show, since normally these functions should not even be referring to the same t . I guess they are showing that under the assumption that $K(t)$ and $r(t)$ start at the same time, and $r(t)$ is determined entirely by the initial distribution of $K(t)$ (without checking whether these assumptions are internally consistent for a particular time series $r(t)$, then (S7) must hold).

The authors need to clearly specify a conceptual model for how K , as a function of time delay t_1 , changes with calendar time t . This will include an anchor specification (meaning a version of S2 with either $K(t-t_1; t)$, or $K(t-t_1; t_1)$).

The authors also need to be more clear about their offsets and how these might change. I note that θ_s is explained tersely after (1) with a reference to F1A, but only θ_p appears in F1A.

Reviewer #3:

Remarks to the Author:

I am happy with the response to my earlier comments

Reviewer #4:

Remarks to the Author:

Quantifying and assessing the impact of the intervention and containment strategies implemented against COVID-19 are challenging and important, perhaps the best time to take up such study to inform the public health management for ongoing pandemic as well as to handle upcoming hazards in advance.

The authors took up this extensive modeling study based on the established hypothesis of the transmission capability of an infectious individual around his/her symptom onset. The authors mainly calibrated the distributions of mainly two related epidemiological parameters: incubation period and serial intervals.

I anticipate that authors will be further encouraged to explain or address the following issues, in particular, to improve the manuscript and its understanding to a broad readership and quantifying the claims.

Major Issues:

1. The authors mentioned the calibration of the epidemiological parameters including serial interval and incubation period were performed during the early stage of the pandemic. It is obvious that the modeling framework is highly based on the initial parameters including the exponential growth rate and basic reproduction number. Where the interventions and containment strategies are temporal in nature, with their time-varying impact on the transmission of the COVID-19. The authors should clarify and discuss this and the inter-relation of the parameters as mentioned in the equation (1). Accordingly, suggest to revise the line in the abstract: "The model is calibrated against incubation period and serial interval statistics during early stages of the pandemic." Further, clarify and specify the rationale of the parameters of the mean reproduction number and mean reproduction rate with their temporal nature for the readers.

2. Continuation of the above point, the serial intervals are not constant over time, in fact, shortened over calendar time (Ali, ST et al. 2020). The authors have cited this article in the reference list but not included in the text. Which had the clear evidence that in mainland China the serial intervals are shortened over time due to the effects of non-pharmaceutical interventions (NPIs), specially contact tracing and isolation of the infectious individuals. The am not sure, the current framework is not accounted for such temporal factors as the calibrations have been done at the initial phase of the outbreaks. Need to be discussed in the text otherwise as one of the limitations of the study.

- Ali, S. T., et al. (2020). "Serial interval of SARS-CoV-2 was shortened over time by nonpharmaceutical interventions." *Science* 369(6507): 1106-1109.

3. The authors estimated the exponential growth rate as 0.3, which seems a bit higher. I wonder authors have overestimated it! I would suggest recheck it, which is one of the prime parameters under this study design. In text it is not clear enough how they have calculated from the data. Only fitting the log-transformed case data always has a risk of overestimation, which should be taken care of. The estimation of the exponential growth rate depends on the length of the exponential phase. Therefore, should be estimated simultaneously to reduce the risk of the overfitting as suggested by Feiver, C., et al. (2006).

- Favier, C., et al. (2006). "Early determination of the reproductive number for vector-borne diseases: the case of dengue in Brazil." *Trop Med Int Health* 11(3): 332-340.

Minor Issues:

1. Line 4-5 in the first paragraph of introduction: update the statistics on COVID-19 as it is presented over two months earlier.

2. Suggest to mention the specific location of data used for the study in the abstract itself.

Response to Reviewer #2

The foundational part (3) seems just wrong to me. The authors frame their model as a renewal equation with kernel K , and then show in the supp that K can be calculated if $r(t)$ is known. This seems practically wrong: in particular, if there is a regime where $r(t)$ is constant, that should be consistent with a broad range of kernel functions, but (3) seems to show that a constant $r(t)$ produces a constant kernel! There is also a weird conflation, since time in $r(t)$ is calendar time, but the primary t in the kernel function is a delay time; these values should not be directly comparable without an offset.

S5 seems just wrong, since the last line seems to show that $r(t)$ is constitutively positive (I guess is missing a term of α). I think this problem is fixed by S7, but it cost me a bunch of time.

I guess S7 is right in some universe of assumptions not clearly laid out by the authors. They are showing that there is some world where $K(t)$ can be inferred from $r(t)$. As mentioned above, this is an unnatural thing to show, since normally these functions should not even be referring to the same t . I guess they are showing that under the assumption that $K(t)$ and $r(t)$ start at the same time, and $r(t)$ is determined entirely by the initial distribution of $K(t)$ (without checking whether these assumptions are internally consistent for a particular time series $r(t)$, then (S7) must hold).

The authors need to clearly specify a conceptual model for how K , as a function of time delay t_1 , changes with calendar time t . This will include an anchor specification (meaning a version of S2 with either $K(t-t_1; t)$, or $K(t-t_1; t_1)$).

Response:

We appreciate the above observations and comments by Reviewer #2 which prompted us to further examine the mathematical structure of our model as expressed by Eqs. (2) and (3). We have carefully reviewed the material in Sec. 1 of the previous version of the SI and checked that all equations presented are correct. Nevertheless, we feel that the simplicity brought about by the constancy of the parameters α_{\square} and β_{\square} could be emphasized a bit more so that our use of a single argument for the kernel function $K(t)$ comes across more readily to the reader.

To achieve the above, we have substantially rewritten Sec. 1.2 (previously Sec. 1.3) of SI. Starting from the second line below Eq. (S6), we give a physical explanation of the two terms in $K(t)$. It is then straightforward to write down Eq. (S7) with the help of the Dirac delta-function. In a similar vein, we arrive at Eq. (S8) for $r(t)$. In each case, we clearly define the beginning and the end of the time interval under consideration. As there is no need to keep track of the temporal

profile of individuals inside the A_1 phase, we hope that we have resolved the queries from Reviewer #2 with the additional information provided.

The authors also need to be more clear about their offsets and how these might change. I note that θ_s is explained tersely after (1) with a reference to F1A, but only θ_p appears in F1A.

Response:

We thank Reviewer #2 for pointing out the error in the Main Text regarding θ_s and Fig. 1a, which has now been corrected. We have modified the right panel of Fig. 1a to illustrate the meaning of θ_s .

Response to Reviewer #3

I am happy with the response to my earlier comments.

We thank Reviewer #3 for going through our manuscript for the second time and for his/her positive assessment of the revised version.

Response to Reviewer #4

General Comments

Quantifying and assessing the impact of the intervention and containment strategies implemented against COVID-19 are challenging and important, perhaps the best time to take up such study to inform the public health management for ongoing pandemic as well as to handle upcoming hazards in advance.

The authors took up this extensive modeling study based on the established hypothesis of the transmission capability of an infectious individual around his/her symptom onset. The authors mainly calibrated the distributions of mainly two related epidemiological parameters: incubation period and serial intervals.

I anticipate that authors will be further encouraged to explain or address the following issues, in particular, to improve the manuscript and its understanding to a broad readership and quantifying the claims.

Response:

We thank the reviewer for these encouragements. We also thank him/her for the suggestions to improve manuscript presentation particularly with regard to the data source used, and for strengthening the comparison of model predictions against the epidemic data.

Specific Comments

Major Issues:

1. The authors mentioned the calibration of the epidemiological parameters including serial interval and incubation period were performed during the early stage of the pandemic. It is obvious that the modeling framework is highly based on the initial parameters including the exponential growth rate and basic reproduction number. Where the interventions and containment strategies are temporal in nature, with their time-varying impact on the transmission of the COVID-19. The authors should clarify and discuss this and the inter-relation of the parameters as mentioned in the equation (1). Accordingly, suggest to revise the line in the abstract: "The model is calibrated against incubation period and serial interval statistics during early stages of the pandemic." Further, clarify and specify the rationale of the parameters of the mean reproduction number and mean reproduction rate with their temporal nature for the readers.

Response:

We thank Reviewer #4 for bringing up this important point. Indeed, our basic model setup Eqs. (1)-(3) is for epidemic outbreaks each with a set of constant epidemiological parameters. While the incubation period distribution $p_o(t)$ is believed to be stable over time and location, the transmission parameters θ_S and α_A , which are derived from the infectiousness function $p_I(t)$, can have a certain level of variability especially when the data set used for calibration contains a significant fraction of imported cases, or is affected by major intervention and containment measures. In this work, we used the transmission pairs compiled by He *et al.* (Ref. 7) to estimate $p_{\square}(t)$. Most of the cases in the data set are from locations outside mainland China, and during the early days of the local outbreaks. Therefore, there is reason to take our estimated $p_{\square}(t)$ as an unbiased representation of transmissibility against infection time t . This view is further supported by the discussion below on serial interval statistics before and after the Wuhan lockdown.

Following the reviewer's suggestion, we have revised the line in the abstract regarding the data used for model calibration to:

“The model is calibrated against incubation period and pairwise transmission statistics during the initial outbreaks of the pandemic outside Wuhan with minimal nonpharmaceutical interventions.”

We have also revised the data description in the subsection on the estimation of the infectiousness function in the Main Text to:

“A data set of 77 pairwise transmissions in several eastern and southeastern Asian countries and regions during their initial COVID-19 outbreak was compiled by He et al.”

The temporal nature of transmission patterns is emphasized in the subsection on serial interval statistics and in the section on epidemic development in various countries and regions.

2. Continuation of the above point, the serial intervals are not constant over time, in fact, shortened over calendar time (Ali, ST et al. 2020). The authors have cited this article in the reference list but not included in the text. Which had the clear evidence that in mainland China the serial intervals are shortened over time due to the effects of non-pharmaceutical interventions (NPIs), specially contact tracing and isolation of the infectious individuals. The am not sure, the current framework is not accounted for such temporal factors as the calibrations have been done at the initial phase of the outbreaks. Need to be discussed in the text otherwise as one of the limitations of the study.

- Ali, S. T., et al. (2020). "Serial interval of SARS-CoV-2 was shortened over time by nonpharmaceutical interventions." *Science* 369(6507): 1106-1109.

Response:

We thank Review #4 for raising the question regarding the significant shift of the serial interval statistics in the five weeks around the Wuhan lockdown, as reported in detail in the paper by Ali et al. Following the reviewer's suggestion, we substantially revised the subsection on serial interval statistics in the Main Text, together with a revised Fig. 2c. We would like to emphasize that the serial interval data were not used in the calibration of our model parameters.

Nevertheless, as Reviewer #4 correctly pointed out, they serve as a benchmark to assess our model predictions. To this effect, we have added the following lines to the Main Text:

“While the overall agreement with the unstratified data is good especially on the positive side, it is also evident that serial intervals can be affected by factors such as the percentage of imported cases, the length of isolation delays, etc. which changed substantially before and after the Wuhan lockdown. As suggested in Ref. 23, their effect can be simulated with a shape function that masks $p_I(t)$. For example, an imported case spent part of his/her infectious period outside the region where the data was collected, shifting $p_{SI}(t)$ to the right. On the other hand, vigorous contact tracing shortens isolation delays significantly, which in turn shifts $p_{SI}(t)$ to the left.”

3. The authors estimated the exponential growth rate as 0.3, which seems a bit higher. I wonder authors have overestimated it! I would suggest recheck it, which is one of the prime parameters under this study design. In text it is not clear enough how they have calculated from the data. Only fitting the log-transformed case data always has a risk of overestimation, which should be taken care of. The estimation of the exponential growth rate depends on the length of the exponential phase. Therefore, should be estimated simultaneously to reduce the risk of the overfitting as suggested by Feiver, C., et al. (2006).

- Favier, C., et al. (2006). "Early determination of the reproductive number for vector-borne diseases: the case of dengue in Brazil." *Trop Med Int Health* 11(3): 332-340.

Response:

We thank the reviewer for raising this issue as well. We understand that estimation of the epidemic growth rate for disease spreading is a task that requires a lot of care. The national/regional figures are further complicated by the possibility of multiple outbreaks in different locations with different starting points and different growth rates, not to mention issues

related to testing and reporting. So there is a limited precision we can attach to the growth rate and methods to estimate it even in the best of circumstances.

Following the reviewer's suggestion, we looked into several schemes to extract the exponential growth rate using the reported daily case numbers. Results shown in the new Fig. 6d is from the simplest scheme we used, where the growth rate on a given day is simply obtained from the local slope of the $\ln N(t)$ versus t curve, using a time interval of three days. As seen from the plot, each data set contains a plateau region of nearly constant growth rate after the cumulative case number reached a few tens or a few hundreds. Beyond that, the growth rate starts to decrease, presumably as a result of intervention and containment measures (but could also due to spreading to other communities with different economic and social settings). This gives us an overall idea of how one might assign a window in either cumulative case number or in time to perform data fitting in each case.

A new section 5 "Estimation of exponential growth rates during initial outbreaks" is added to SI to explain the above issues in some detail and also to present the estimated COVID-19 growth rates during the early stage of the 1st wave in various countries/regions following the scheme proposed by Favier *et al.* In some cases such as Italy, one indeed obtains a lower value of the exponential growth rate using the Favier *et al.* scheme to select the window to perform data fitting. This, together with Fig. 6d and the additional references 38 and 39 in the revised Main Text, should allow the reader to obtain a more complete view of similarities and differences of the COVID-19 outbreaks across countries and continents.

Minor Issues:

1. Line 4-5 in the first paragraph of introduction: update the statistics on COVID-19 as it is presented over two months earlier.

We thank the reviewer for highlighting this and have updated the statistics of COVID-19.

2. Suggest to mention the specific location of data used for the study in the abstract itself.

We thank the reviewer for this suggestion and have updated the relevant section, as addressed in our response to Question 1.

Finally, we thank Reviewer #4 again for her/his insightful and constructive comments. We hope our responses above have addressed these important issues/concerns in a satisfactory manner.

Reviewers' Comments:

Reviewer #2:

Remarks to the Author:

My comments have not been addressed nor (apparently) understood.

If the authors' mathematical results are indeed correct, they apparently apply to some model world which they have not clearly described, nor clearly connected to the real world: K cannot in general be inferred from $r(t)$.

I don't believe there's a clear _scientific_ story here, just a mathematical result behind a tangle of poorly justified scientific assumptions. If there _is_ a clear story here, it is not sufficiently explained to be followed by more than a tiny fraction of the likely readership.

Reviewer #4:

Remarks to the Author:

Yes, I can see the authors have sufficiently revised the manuscript incorporating the comments and suggestions of the reviewers.

Clarification:

I think the authors mean by the line (page 5),

"As suggested in Ref. 23, their effect can be simulated with a shape function that masks $p_1(t)$." is as below and prefer to revise the sentence as:

"Therefore, such temporal effect on the serial interval can be simulated simply with a shape function that masks $p_1(t)$."?

Reviewer #5:

Remarks to the Author:

As far as I can see, the authors deal with a special case of the linearized version of the general Kermack-McKendrick model from 1927, see the original paper and <https://www.tandfonline.com/doi/full/10.1080/17513758.2012.716454>
The references [17,18] do not elucidate this.

The authors call the kernel of the linear renewal equation $r(t)$.

The use of the symbol t as argument of r is unfortunate, as in their second equation they use t to denote real/absolute time. I strongly recommend to systematically use a different character to denote the 'time elapsed since infection took place'. The present choice of notation is bound to create confusion (also r is dangerous, as usually it denotes the Malthusian parameter).

The key idea of the paper seems to be to relate r to the distribution of symptom onset and to use data about the latter to infer the first. But exactly how remains mysterious to me : what is the basis for (1) and how is θ_S defined, if all you have are observations of symptom onset ? And what exactly motivates the more detailed stochastic model that you choose to deduce expressions for r ? What are the advantages ? As the detailed model still contains the functions α_L and β_B as ingredients, you remain in the area of nonparametric statistics, see <https://www.cambridge.org/core/books/nonparametric-estimation-under-shape-constraints/881B662EEF5B5266E5E4D80E6153FCDA> for possibly useful methodology. Are you choosing the more detailed model in order to facilitate

the incorporation of control measures ?

To summarize : in my opinion the assumptions are neither clearly described nor well motivated. To convince, the presentation should be much improved.

Admittedly, I did not read the entire manuscript and I do have little experience with data analysis. But I do have a lot of experience in epidemic modeling, and the opinion given above is based on a serious study of pages 1-3 of the main text and pages 3-5 of the supp.

A detail : in the middle line of both S2 and S5 the integration symbol dt_1 is misplaced.

Reviewer #6:

Remarks to the Author:

It makes sense to me that this model should be as tractable as an elaborated compartmental model SEI1I2I3R because of the rather strong assumptions:

- Phase I1 has constant exit rate and constant transmissibility.
- Phase I2 has constant duration and transmissibility that is linearly decreasing with time
- Phase I3 has a transmission rate that decays exponentially at a high rate, which is well-represented as a pulse of transmissions on entry.

Overall, then, the expected number of transmissions for a person entering phase I1 is easily determined. Since everyone who is infected passes through each of these compartments in sequence, the reproductive number $r(t)$ should not be difficult to determine.

I agree with Reviewer 2 that the kernel derivation (S3 to S5) could be made clearer, explicitly showing the changes of variable and especially the interchanged limits of integration. Using the dummy variable t as an argument for the kernel in Eqs. S3 and S7 is confusing, since in use it is replaced by the time interval $t-t_1$, where " t " has a very different meaning. I think this is the source of the reviewer's request for an "anchor" time argument. The kernel itself is independent of any anchor time – that dependence comes in through its convolution with $A_1(t)$ – but the notation makes it seem like it depends only on the anchor time. Simply switching to a dummy argument of Δ might reduce some of the understandable confusion expressed by the reviewer, which is likely to arise in many readers' minds.

Also, in Eq. S2, the integral over t_1 must include the second factor, since it also depends on t_1 . I.e., move " dt_1 " to the end of the expression. And the θ parameters should be defined in the text, not just in figures, before their otherwise abrupt introduction in Eq. 1.

However, in my opinion the model is technically valid – I don't understand the reviewer's other comments, at least in regard to the current version.

- The reproductive number $r(t)$ is constitutively positive: the number of transmissions per person cannot be negative. $r(t)$ is the exponential of a rate that is sometimes also called the transmission rate, which may obviously be positive or negative.
- a constant $r(t)$ does not require a constant kernel. Eq. S5 clearly shows this is not the case. Indeed, if $r(t)$ is constant, $K(t-t_1)$ must be a delta-function, which makes sense.
- The limits of integration show that both $r(t)$ and the kernel are assumed to "start" at time $t=0$, though a better way of putting it might simply be that $A_1(t)$ is 0 before $t=0$. It is not unusual to assume the distribution of infection times starts with a pulse at $t=0$.
- The model itself is separate from any time-series. If "these assumptions are [not] consistent for a particular time series" it does not mean the model itself is inconsistent, only that it may be a poor representation of reality.

I note that, as agreed with the editor, the scope of this review is confined to the above details of the model itself.

Point-to-point response to reviewers' comments and suggestions

Reviewer #2

My comments have not been addressed nor (apparently) understood.

If the authors' mathematical results are indeed correct, they apparently apply to some model world which they have not clearly described, nor clearly connected to the real world: K cannot in general be inferred from $r(t)$.

I don't believe there's a clear scientific story here, just a mathematical result behind a tangle of poorly justified scientific assumptions. If there is a clear story here, it is not sufficiently explained to be followed by more than a tiny fraction of the likely readership.

Response:

We thank Reviewer #2 for taking yet another look at our manuscript, and sincerely apologise for missing the key part of his criticism regarding notation and model presentation in the last round of revision. With the helpful inputs from Reviewer #5 and #6, we have thoroughly revised the model description section in the Main Text and the exposition of the governing equations for epidemic development in a large, well-mixed community in SI, Secs. 1 and 3. The calendar time, now denoted by capital letter T , is singled out clearly from elapsed times and time constants throughout the Main Text and SI.

Equations (2) and (3) in the previous version of the manuscript, which generated much concern from Reviewer #2, are technically correct but turn out to be an unnecessary detour to the results presented in this work. We would like to invite Reviewer #2 to go through the revamped formulation in Sec. 1 of SI, and are hopeful that the scientific story is now more clearly articulated.

Reviewer #4

Yes, I can see the authors have sufficiently revised the manuscript incorporating the comments and suggestions of the reviewers.

Clarification:

I think the authors mean by the line (page 5), “As suggested in Ref. 23, their effect can be simulated with a shape function that masks $p_1(t)$.” is as below and prefer to revise the sentence as: “Therefore, such temporal effect on the serial interval can be simulated simply with a shape function that masks $p_1(t)$.”?

Response:

We are grateful to Reviewer #4 for the many helpful comments and suggestions in the previous round which contributed greatly to the improvement of the paper. The suggested clarification above has been incorporated in the latest revision.

Reviewer #5

As far as I can see, the authors deal with a special case of the linearized version of the general Kermack-McKendrick model from 1927, see the original paper and <https://www.tandfonline.com/doi/full/10.1080/17513758.2012.716454>
The references [17,18] do not elucidate this.

Response:

We thank Reviewer #5 for introducing to us the seminal work by Kermack and McKendrick and subsequent developments and we are delighted to see that our current study has a large overlap with the established school of thought. The papers are now cited as the first two references in SI.

The authors call the kernel of the linear renewal equation $r(t)$. The use of the symbol t as argument of r is unfortunate, as in their second equation they use t to denote real/absolute time. I strongly recommend to systematically use a different character to denote the ‘time elapsed since infection took place’. The present choice of notation is bound to create confusion (also r is dangerous, as usually it denotes the Malthusian parameter).

Response:

We thank Reviewer #5 for the comment and suggestion. His remark to link $r(t)$ with the kernel of the renewal equation prompted us to re-examine Eqs. (2) and (3) in the previous version of the manuscript which are now replaced by two different equations. This reformulation, together with the suggested notational change, lend much simplicity and clarity to the mathematical treatment in Sec. 1 of SI. As a result, the amount of work required for the reader to go through the material in the section is much reduced.

With regard to the usage of $r(t)$ for the mean reproduction rate on day t since infection, we feel that the chance of being misunderstood is slim and also we are afraid that, with the large number of symbols already in use in the paper, the choice of alternatives is rather limited. Therefore, we have kept it intact.

The key idea of the paper seems to be to relate r to the distribution of symptom onset and to use data about the latter to infer the first. But exactly how remains mysterious to me : what is the basis for (1) and how is θ_S defined, if all you have are observations of symptom onset ? And what exactly motivates the more detailed stochastic model that you choose to deduce expressions for r ? What are the advantages ?

Response:

Indeed, the main objective of the paper is to relate the statistics of pair-wise transmission in individual case studies to epidemic development at the population level. Within the linear framework, $r(t)$ plays the dual role of the rate of reproduction at the individual level, which is stochastic, and the kernel function of the renewal equation for the size of infected (sub)populations, which is deterministic. However, direct determination of $r(t)$ from clinical data is difficult as this would require complete transmission history from a large number of patients. The compartmentalised, stochastic model allows one to break up this task into two steps as illustrated in Fig. 1a of the Main Text, taking advantage of the transmission characteristics of COVID-19.

Eq. (1) is both a representation of empirical observations and also an outcome of the compartmentalised model. Given the overwhelming clinical evidence that infectiousness of a COVID-19 patient peaks around the symptom onset, the proposed expression for $r(t)$ is quite plausible. Further to the empirical observation, Sec. 3.1 in SI presents a derivation of Eq. (1) from the compartmentalised model. An expression for the phenomenological parameter θ_s is given and its numerical value is estimated in Sec. 2 from clinical data.

As the detailed model still contains the functions α_L and β_B as ingredients, you remain in the area of nonparametric statistics, see <https://www.cambridge.org/core/books/nonparametric-estimation-under-shape-constraints/881B662EEF5B5266E5E4D80E6153FCDA> for possibly useful methodology. Are you choosing the more detailed model in order to facilitate the incorporation of control measures ?

Response:

The compartmentalised model is introduced to simulate COVID-19 disease development and transmission at the individual level, including a quantitative determination of $r(t)$ from clinical data. Reviewer #5 is right that, in the definition of the model, we leave the functional forms for $\alpha_L(t)$ and $\beta_B(t_B)$ free so as to be able to produce better agreement between the model and clinical data. The procedure to determine the two functions is discussed at length in the Methods section and also in Sec. 2 of SI. While the disease transmission data suggests that $\beta_B(t_B)$ is well described by a simple exponential function with a decay constant α_B , the temporal structure of $\alpha_L(t)$ is a bit more complex in order to accommodate the key statistical characteristic of the latent phase. We added a new subsection 2.3 in SI to present our observations.

Upon careful calibration, the compartmentalised model offers a convenient starting point to quantitatively assess various intervention measures, as described in the Main Text and SI.

To summarize : in my opinion the assumptions are neither clearly described nor well motivated. To convince, the presentation should be much improved.

Admittedly, I did not read the entire manuscript and I do have little experience with data analysis. But I do have a lot of experience in epidemic modeling, and the opinion given above is based on a serious study of pages 1-3 of the main text and pages 3-5 of the supp.

Response:

We much appreciate the constructive advice and many useful suggestions from Reviewer #5, which prompted us to perform an overhaul of the model formulation and the key equations (2) and (3) in the Main Text. Section 1.1 in SI is completely rewritten along the lines suggested by the reviewer. The revised manuscript should be more accessible to the reader, both in the flow of ideas and the communication of results obtained, and for this we are grateful to Reviewer #5.

A detail : in the middle line of both S2 and S5 the integration symbol dt is misplaced.

Response:

We have followed Reviewer #5's advice and rewrote much of the mathematical derivations in Sec. 1.1, SI.

Reviewer #6

It makes sense to me that this model should be as tractable as an elaborated compartmental model SEI1I2I3R because of the rather strong assumptions:

- Phase I1 has constant exit rate and constant transmissibility.
- Phase I2 has constant duration and transmissibility that is linearly decreasing with time
- Phase I3 has a transmission rate that decays exponentially at a high rate, which is well-represented as a pulse of transmissions on entry.

Overall, then, the expected number of transmissions for a person entering phase I1 is easily determined. Since everyone who is infected passes through each of these compartments in sequence, the reproductive number $r(t)$ should not be difficult to determine.

Response:

We thank Review #6 for his detailed and positive summary of our model.

I agree with Reviewer 2 that the kernel derivation (S3 to S5) could be made clearer, explicitly showing the changes of variable and especially the interchanged limits of integration. Using the dummy variable t as an argument for the kernel in Eqs. S3 and S7 is confusing, since in use it is replaced by the time interval $t-t_1$, where “ t ” has a very different meaning. I think this is the source of the reviewer’s request for an “anchor” time argument. The kernel itself is independent of any anchor time – that dependence comes in through its convolution with $A1(t)$ – but the notation makes it seem like it depends only on the anchor time. Simply switching to a dummy argument of Δt might reduce some of the understandable confusion expressed by the reviewer, which is likely to arise in many readers’ minds.

Response:

We thank Reviewer #6 for emphasising the importance of distinguishing the calendar time from the elapsed times, originally brought up by Reviewer #2. We have followed the advice of all three reviewers and thoroughly changed the notation, using the capital letter T for calendar days and the lowercase t for elapsed times, both in the Main Text and SI. Additional notational changes were made for consistency in the usage of “time from the symptom onset” Δt and the serial interval t_{SI} . With these revisions, we hope the possible confusions are much reduced.

Also, in Eq. S2, the integral over $t1$ must include the second factor, since it also depends on $t1$. I.e., move “ $dt1$ ” to the end of the expression. And the θ parameters should be defined in the text, not just in figures, before their otherwise abrupt introduction in Eq. 1.

Response:

We thank Review #6 for these suggestions which have been implemented in the revised version.

However, in my opinion the model is technically valid – I don't understand the reviewer's other comments, at least in regard to the current version.

- The reproductive number $r(t)$ is constitutively positive: the number of transmissions per person cannot be negative. $r(t)$ is the exponential of a rate that is sometimes also called the transmission rate, which may obviously be positive or negative.
- a constant $r(t)$ does not require a constant kernel. Eq. S5 clearly shows this is not the case. Indeed, if $r(t)$ is constant, $K(t-t_1)$ must be a delta-function, which makes sense.
- The limits of integration show that both $r(t)$ and the kernel are assumed to “start” at time $t=0$, though a better way of putting it might simply be that $A_1(t)$ is 0 before $t=0$. It is not unusual to assume the distribution of infection times starts with a pulse at $t=0$.
- The model itself is separate from any time-series. If “these assumptions are [not] consistent for a particular time series” it does not mean the model itself is inconsistent, only that it may be a poor representation of reality.

I note that, as agreed with the editor, the scope of this review is confined to the above details of the model itself.

Response:

We thank Reviewer #6 for carefully going through our work and for his/her factual and positive assessments. While the original mathematical formulation, with an emphasis on the size of the infected population in the A_1 phase, is technically sound, we have since discovered that one can do away with the somewhat awkward kernel $K(t)$ which was the original source of objection from Reviewer #2. Upon advice from all three reviewers, we have substantially revised Sec. 1.1 of SI and made corresponding changes to the rest of the manuscript, including the new equations (2) and (3) in the Main Text. We much appreciate Reviewer #6's thorough understanding of our work.

Reviewers' Comments:

Reviewer #5:

Remarks to the Author:

In my opinion the manuscript is now clear enough to be accepted.

Reviewer #6:

Remarks to the Author:

My concerns with the previous version have been addressed well. I think the result is a description of a model that will be of widespread interest because it relates two established approaches. I have again confined my review to the model description.

It never fails to amaze me how stove-piped we have become – in this case, developing an epidemic model without having encountered Kermack-McKendrick. If they have not already, the authors should study the work of Heesterbeek, Dietz, and Diekmann (e.g.,

Heesterbeek J.A.P, Dietz K. The concept of R_0 in epidemic theory. Stat. Neerl. 1996;50:89–110

DIEKMANN, O., J. A. P. HEESTERBEEK and J. A. J. Metz (1990), On the definition and the computation of the basic reproduction ratio R_0 in models for infectious diseases in heterogeneous populations, Journal of Mathematical Biology 28, 365-382.)

Their work reviews the historical development of R_0 , its application to structured populations and, if memory serves, its extension to non-Poisson transition rates.

Response to Reviewer #5

In my opinion the manuscript is now clear enough to be accepted.

We thank Reviewer #5 for reviewing our paper again. We are pleased to know that s/he is satisfied with the revised version.

Response to Reviewer #6

My concerns with the previous version have been addressed well. I think the result is a description of a model that will be of widespread interest because it relates two established approaches. I have again confined my review to the model description.

We thank Reviewer #6 for reviewing our paper again and her/his enthusiastic assessment on our model. We are pleased to know that s/he is satisfied with the revised version.

It never fails to amaze me how stove-piped we have become – in this case, developing an epidemic model without having encountered Kermack-McKendrick. If they have not already, the authors should study the work of Heesterbeek, Dietz, and Diekmann (e.g.,

Heesterbeek J.A.P, Dietz K. The concept of R_0 in epidemic theory. Stat. Neerl. 1996;50:89–110

DIEKMANN, O., J. A. P. HEESTERBEEK and J. A. J. Metz (1990), On the definition and the computation of the basic reproduction ratio & in models for infectious diseases in heterogeneous populations, Journal of Mathematical Biology 28, 365-382.)

Their work reviews the historical development of R_0 , its application to structured populations and, if memory serves, its extension to non-Poisson transition rates.

We thank Reviewer #6 for bringing these classical works to our attention. A short paragraph is added at the beginning of the Supplementary Information where the historical developments were referenced.